# Fine-tuning with Reserved Majority for Noise Reduction

**Shuyang Jiang**[♠,♣]**, Yusheng Liao**[◇,♣]**, Ya Zhang**[◇,♣,*]**, Yanfeng Wang**[◇,♣]**, Yu Wang**[◇,♣,*]
[♠]Fudan University
[◇]School of Artificial Intelligence, Shanghai Jiao Tong University
[♣]Shanghai Artificial Intelligence Laboratory
`shuyangjiang23@m.fudan.edu.cn`
`{liao20160907,ya_zhang, wangyanfeng622, yuwangsjtu}@sjtu.edu.cn`

## Abstract

Parameter-efficient fine-tuning (PEFT) has revolutionized supervised fine-tuning, where LoRA and its variants gain the most popularity due to their low training costs and zero inference latency. However, LoRA tuning not only injects knowledgeable features but also noisy hallucination during fine-tuning, which hinders the utilization of tunable parameters with the increasing LoRA rank. In this work, we first investigate in-depth the redundancies among LoRA parameters with substantial empirical studies. Aiming to resemble the learning capacity of high ranks from the findings, we set up a new fine-tuning framework, **P**arameter-**Re**dundant **F**ine-**T**uning (PRₑFT), which follows the vanilla LoRA tuning process but is required to reduce redundancies before merging LoRA parameters back to pre-trained models. Based on this framework, we propose **No**ise reduction with **R**eserved **M**ajority (NₒRM), which decomposes the LoRA parameters into majority parts and redundant parts with random singular value decomposition. The major components are determined by the proposed ***Sim-Search*** method, specifically employing subspace similarity to confirm the parameter groups that share the highest similarity with the base weight. By employing NₒRM, we enhance both the learning capacity and benefits from larger ranks, which consistently outperforms both LoRA and other PRₑFT-based methods on various downstream tasks, such as general instruction tuning, math reasoning and code generation. Code is available at `https://github.com/pixas/NoRM`.

## 1 Introduction

Large language models (LLMs) have revolutionized Natural Language Processing (NLP) by pretraining on vast textual corpora, enabling them to encode extensive world knowledge (Chang et al., 2024; AlKhamissi et al., 2022). These models exhibit remarkable zero-shot and few-shot performance across a wide range of tasks (Brown et al., 2020; Anil et al., 2023; OpenAI, 2023; Touvron et al., 2023a;b). Instruction tuning, a form of supervised fine-tuning (SFT), has further enhanced LLMs by refining their instruction-following capabilities, thereby simplifying human-LLM interactions (Ouyang et al., 2022; Chung et al., 2024). Pretrained models fine-tuned with SFT excel in various downstream tasks such as medical consultation (Wu et al., 2024; Chen et al., 2023), mathematical reasoning (Luo et al., 2023a; Yue et al., 2023; Yu et al., 2024b; Toshniwal et al., 2024), and serving as intelligent assistants (Peng et al., 2023; Ivison et al., 2023). However, adapting these models to downstream tasks in resource-constrained environments requires parameter-efficient fine-tuning (PEFT) techniques, which update less than 1% of the total parameters while achieving performance comparable to full fine-tuning. LoRA and its extensions (Pfeiffer et al., 2021; yang Liu et al., 2024; Hayou et al., 2024; Wang et al., 2024; Jiang et al., 2024b) have emerged as some of the most effective and efficient methods in this regard.

Despite its efficiency, vanilla LoRA fine-tuning does not always benefit from increasing the number of tunable parameters and may even degrade performance. Hu et al. (2022) found that optimal

---

*Corresponding Author

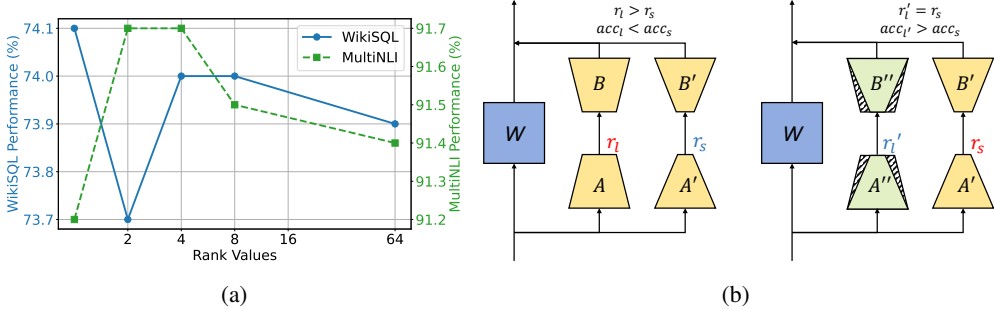

(a)                                                                                                   (b)

Figure 1: Vanilla LoRA tuning usually cannot benefits from larger ranks **(a)** and **Left of (b)**, where $r$ represents the rank of LoRA and $acc$ represents the downstream performance. In contrast, PREFT aims to adopt the same number of tunable parameters as LoRA but cuts part of it to achieve higher performance **(Right of (b))**, where $r'_l < r_s$ and $r_l > r_s$.

performance is often achieved with lower ranks, while larger ranks lead to negligible or negative improvements (see Figure 1a). We hypothesize that this behavior stems from the fine-tuning dynamics of LLMs. As shown by Gekhman et al. (2024), fine-tuning can introduce both useful downstream knowledge and undesired hallucinatory features. We speculate that as the number of tunable parameters in LoRA modules increases, fine-tuning results in a trade-off between valuable knowledge and noisy features. Consequently, increasing the rank may exacerbate the hallucination problem, leading to diminished performance gains. Thus, improving LoRA's effectiveness requires selectively retaining useful knowledge while eliminating unwanted redundancies.

In this paper, we systematically investigate the parameter redundancies in LoRA fine-tuning. Our analysis spans the overall structure of LLMs, focusing on transformer layers and specific modules within the transformer. Our experiments reveal that LoRA parameters contain significant redundancies, which exhibit distinct patterns across layers and modules. Based on these findings, we propose a new parameter-efficient tuning framework, **P**arameter **R**edundancies **F**ine-**T**uning (PREFT), which follows the standard LoRA tuning pipeline during training but eliminates certain redundancies before merging the fine-tuned parameters back into the base LLMs (see Figure 1b). To further optimize this process, we introduce **No**ise reduction with **R**eserved **M**ajority (NORM), which utilizes singular value decomposition (SVD) to distinguish between essential and noisy components. NORM accelerates the computation by using an approximated SVD and introduces a novel **Sim-Search** method that adaptively selects the most valuable components based on subspace similarity between the reduced and base weights. Figure 3 illustrates the computational flow of NORM and **Sim-Search**.

We conduct comprehensive experiments to validate the effectiveness of NORM, covering tasks such as general instruction tuning, mathematical reasoning, and code generation, using three strong pre-trained models. NORM consistently outperforms LoRA and other PREFT methods, achieving an average gain of **+4.67** over the best PEFT methods and **+1.63** over the strong PREFT method TAIA, when applied to Llama3-8B. Additional analysis confirms the robustness of NORM, and shows that **Sim-Search** outperforms alternative similarity-based search methods. Further experiments demonstrate that NORM significantly improves the utilization of the fine-tuning corpus while maintaining the retention of pre-trained knowledge.

Overall, we conclude our contributions as such:

1. **Revisiting LoRA Fine-Tuning**: We revisit LoRA tuning and, through extensive experiments, reveal that it suffers from low parameter utilization due to the introduction of redundant features. To address this, we introduce PREFT, a novel fine-tuning framework that significantly improves upon existing PEFT methods by reducing these redundancies.

2. **Adaptive Noise Reduction at Inference**: Within the PREFT framework, we propose NORM, an adaptive method that removes disruptive components for each parameter. This is achieved using our automated search algorithm, **Sim-Search**, which identifies the most relevant components by evaluating their affinity with the base weights.

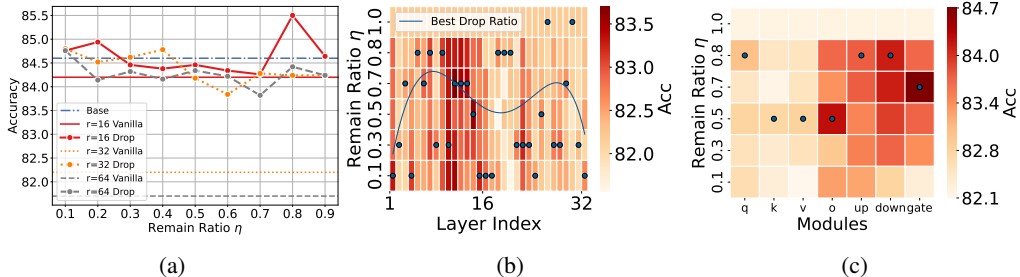

Figure 2: Performance change log with random drop ratios (a) and the performance distribution among layers (b) and modules (c). We annotate the best remaining ratio with darkblue.

3. **Comprehensive Evaluation**: We rigorously evaluate NORM across multiple domains, including general instruction tuning, mathematical reasoning, and code generation. Our method consistently surpasses state-of-the-art PEFT methods and alternative PREFT approaches, demonstrating the effectiveness of removing parameter redundancies.

## 2 PARAMETER REDUNDANCIES FINE-TUNING (PREFT)

In this section, we first introduce the basic properties of LoRA fine-tuning. After that, we empirically unveil the common parameter redundancies within PEFT-based methods. Following the findings, we formalize the post-processing of these redundancies as a novel fine-tuning framework, PREFT. Ultimately, we introduce two genres of identifying the redundant components intelligently.

### 2.1 PRELIMINARIES

Given an input sequence $\mathbf{X} \in \mathbb{R}^{m \times d}$, LoRA (Hu et al., 2022) proves that the update of original linear layers $\Delta W$ in large language models is of low-rank, and can be decomposed into the multiplication of two compact matrices $AB$. The pretrained weight $W \in \mathbb{R}^{d' \times d}$ is frozen in the training phase, while $A$ and $B$ are trainable parameters and contribute together to the forward pass:

$$\mathbf{H} = \mathbf{X}W^\top + \mathbf{X}\Delta W^\top = \mathbf{X}W^\top + \frac{\alpha}{r}\mathbf{X}(BA)^\top \tag{1}$$

where $A \in \mathbb{R}^{r \times d}, B \in \mathbb{R}^{d' \times r}$ and $r \ll d, d'$. $\alpha$ is the scaling factor. Without loss of generality, we omit the layer index for the following formula. At the beginning of training, $B$ is initialized to an all-zero matrix and $A$ uses Gaussian initialization to ensure that $BA$ is zero at initialization.

### 2.2 PILOT EXPERIMENTS

Yu et al. (2024a) indicates that the delta parameters of full fine-tuned models contain superior redundancies, where over ninety percent of fine-tuned parameters can be dropped. In contrast to them, in this study, we further dig out the parameter redundancies among PEFT modules, which are usually convinced of compact encoding compared to full parameters. We progressively present the commonly existing redundancies holistically, along with layer-wise and module-wise analysis.

**Experiment Settings**   We mainly utilize `Llama3-8B-Instruct` (AI@Meta, 2024) as the base model. We choose MetaMathQA-395K (Yu et al., 2024b) as the fine-tuning corpus and SVAMP (Patel et al., 2021) as the evaluation set. We adopt vanilla LoRA (Hu et al., 2022) as the fine-tuning method and choose three configurations of LoRA rank and $\alpha$ values: $\{(r, \alpha) \mid (16, 32), (32, 64), (64, 128)\}$. The learning rate is set to 2e-4 and the total batch size is set to 128. After fine-tuning, we set up three parameter drop strategies: (1) drop holistically, (2) drop by layer, and (3) drop by module.

**Drop holistically:**   We denote the remaining ratio of intrinsic rank as $\eta$. We **randomly** remain $\eta = 10\% \sim 90\%$ channels of both parameters: $A' = A[H, :]$ and $B' = B[:, H]$ where $H = \{h_1, h_2, \ldots, h_{r'} \mid h_i \in \{0, 1, \cdots, r-1\} \land \forall i, j \ (i \neq j) \Rightarrow h_i \neq h_j\}$ and $r' = \lfloor \eta \cdot r \rfloor$. To maintain the scale of modified delta parameters, we also change the $\alpha$ value from $2r$ to $r'$ after the above

modification. Experiments are averaged by five runs to reduce random bias. In Figure 2a, it is obvious that even with a low rank ($r = 16$), randomly dropping brings significant performance gain compared to complete parameters. Meanwhile, a small remaining ratio ($\eta = 0.1$) still brings performance gains, which indicates mass redundancies among each parameter. We also notice that lower ranks bring higher performances, which is attributed to fewer introduced hallucinations (Gekhman et al., 2024) by the intrinsic properties of fine-tuning.

**Drop by layer:** We find that the LoRA rank has no effect on deriving conclusions; therefore, in the following experiments, we only choose $r = 32$ for simplicity. For each layer $l \in [0, L)$ of base LLMs, we select five remaining ratios: $\{10\%, 30\%, 50\%, 70\%, 80\%\}$ as well as the full LoRA parameter as remaining ratio $100\%$ and follow the first strategy to retain delta LoRA parameters of the specific layer yet keep the other layers unchanged. For layer redundancies (Figure 2b), we find that in middle layers, the best performances are achieved with moderate drop ratios. For upper layers and lower layers, a large remaining ratio generally brings higher results. This indicates that initial encoding and eventual output layers contain much fewer redundancies compared to middle layers, which also aligns with previous findings (Men et al., 2024).

**Drop by module:** For each module of a Llama-style transformer, including $\{$`q_proj`,`k_proj`, `v_proj`, `o_proj`, `gate_proj`,`up_proj`,`down_proj`$\}$, we choose five remaining rates $\{10\%, 30\%, 50\%, 70\%, 80\%\}$ and follow the second strategy to retain delta LoRA parameters of the specific module yet keep the other modules unchanged. In Figure 2c, dropping parameters of $\{$`q_proj`,`k_proj`,`v_proj`$\}$ brings minor performance gains while dropping those of $\{$`o_proj`, `gate_proj`, `up_proj`, `down_proj`$\}$ facilitates further performance improvements, which is also alluded to by Jiang et al. (2024a). Noticeably, for the latter four modules, a small drop ratio like 30% or 50% generally results in the best performance across all drop ratios, which also indicates that MLP blocks encode nonnegligible hallucinatory information during PEFT.

## 2.3 PREFT

Built upon these findings, it is essential to adaptively remove these redundancies based on parameter locations to enhance the effectiveness of LoRA parameters. Therefore, we propose a new fine-tuning framework: **P**arameter **R**edundancies **F**ine-**T**uning (PREFT), which enhances fine-tuned models by removing redundancies of PEFT parameters obtained via a usual training procedure. The comparison with PEFT is illustrated in Figure 1b. Formally, given the pretrained weight $W$ and updated $\Delta W$, PREFT aims to create new delta weights $\Delta W'$ [1] by removing parts of $\Delta W$, under the condition that:

$$\arg\max M(\boldsymbol{x} \mid \{W_i\}_{i=1}^{p}, \{\Delta W_i'\}_{i=1}^{p}) \tag{2}$$

where $p$ is the total number of parameters, $\boldsymbol{x}$ is the input sequence and $M$ is a specific metric for downstream tasks. Based on it, previous method TAIA (Jiang et al., 2024a) can be concluded as such:

$$\{\Delta W'\} = \{\Delta W_{attn}, \mathbf{0}_{ffn}\} \tag{3}$$

where it removes all delta parameters of FFN modules but keeps the self-attention part unchanged. Another example, MedCare (Liao et al., 2024), also adopts PREFT philosophy:

$$\{\Delta W'\} = \{\Delta W_{LoRA}, \mathbf{0}_{MoLoRA}\} \tag{4}$$

where $\mathrm{MoLoRA}$ is the mixture-of-LoRA submodule (Feng et al., 2024; Liu et al., 2023a). However, these two instances of PREFT identify the shearing part through empirical observations and lack fine-grained parameter-wise practice for reducing redundancies. Consequently, in the newly introduced PREFT system, the core of adaptive parameter shearing is to identify redundant parts of original LoRA parameters more intelligently. Based on this, we categorize the identification process into two genres: (1) intra-shearing which leverages solely LoRA parameters to perform reduction, and (2) inter-shearing which leverages the relationship between LoRA parameters and corresponding base weights to fulfill the task. We exemplify these two categories in the following:

**Intra-shearing** Assuming that we want to keep $0 < \beta < 1$ components of each parameter ($\beta$ can be searched on the dev set), the most common practice is to perform principle component

---

[1]Without loss of generality, we use $(\cdot)'$ notation for modified parameter under PREFT framework.

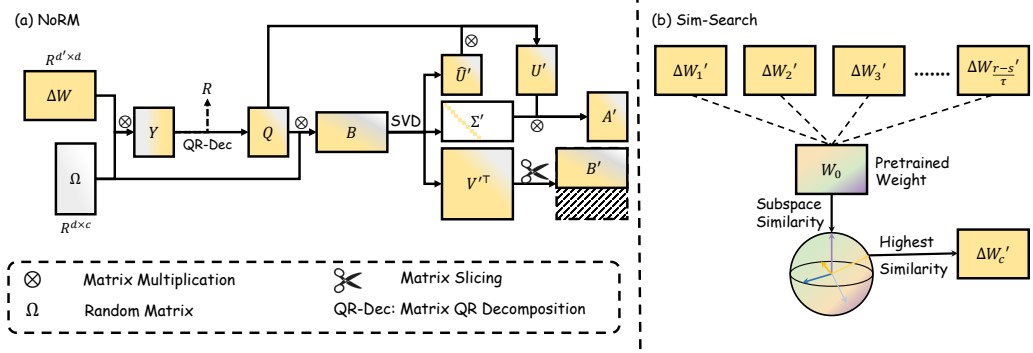

Figure 3: Overview of NORM. NORM employs random SVD to extract the major components from the delta parameter. NORM utilizes (b) **Sim-Search** to determine $c$ channels with little hallucination based on the subspace similarity between the sheared delta weight and the pre-trained weight.

analysis (PCA (Abdi & Williams, 2010)) or singular value decomposition (SVD (Klema & Laub, 1980)) to dig out the major components:

$$A_r, B_r = \kappa(A, B) \tag{5}$$

where $\kappa(\cdot)$ can be any method that selects the major $\beta$ percent of total components.

**Inter-shearing** In the above method, we need to search $\beta$ on the dev set, which inhibits PREFT from the deployment on real applications. Therefore, inter-shearing methods determine $\beta$ on the relationship between LoRA weights and pre-trained weights, which is based on an observation that fine-tuning introduces hallucination sharing different distributions with pre-trained weights (Gekhman et al., 2024). Therefore, we choose $\beta$ such that the remaining components overlap with the pre-trained weight to a maximum extent. Formally, we can conduct a hyperparameter search to determine $\beta$:

$$\beta = \arg\max_{\beta} \mathrm{Sim}(B_\beta A_\beta, W) \tag{6}$$

where $\beta$ can be searched over a given range with a pre-defined step and $\mathrm{Sim}$ is any similarity-based metric between the delta and base weight, including the reverse of $L_2$-distance or cosine similarity.

Based on the above categorization, in this paper, we build an inter-shearing method called **No**ise reduction with **R**eserved **M**ajority (NORM) by reserving contributing components through random SVD while determining $\beta$ by a novel **Sim-Search** method, which maximizes the subspace similarity between the reduced LoRA weights and base weights.

## 3 NORM

### 3.1 OVERALL PIPELINE

The overall pipeline for NORM is presented in Figure 3. We start by analyzing the approximation of LoRA parameters $B \in \mathbb{R}^{d' \times r}$ and $A \in \mathbb{R}^{r \times d}$ and their product $\Delta W = BA$. By the low-rank decomposition property of LoRA, we can get $\mathrm{Rank}(BA) = r$. Without taking any low-rank assumptions on the updated weight $\Delta W$, we decompose it using singular value decomposition (SVD):

$$\Delta W = \mathbf{U}\mathbf{\Sigma}\mathbf{V}^\top \tag{7}$$

where $\mathbf{U} \in \mathbb{R}^{d' \times r}, \mathbf{V} \in \mathbb{R}^{d \times r}$ are left/right singular matrices and $\mathbf{\Sigma} \in \mathbb{R}^{r \times r}$ is the diagonal matrix containing the singular values of $\Delta W$. To approximate the delta parameters by discarding redundant components, we can retain the first $c < r$ largest singular values and corresponding singular vectors:

$$\mathbf{\Sigma}' = \mathrm{diag}(\sigma_1, \sigma_2, \cdots, \sigma_c), \quad \mathbf{U}' = \mathbf{U}[:, 1:c], \quad \mathbf{V}' = \mathbf{V}[:, 1:c] \tag{8}$$

Such approximation deduces the approximation of $BA$: $\Delta W = BA \approx \mathbf{U}'\mathbf{\Sigma}'\mathbf{V}'^\top$. However, directly computing the singular value decomposition of delta weight $\Delta W$ is computationally heavy for both

pre-processing and storage; therefore, we propose to use randomized SVD (Halko et al., 2011) to further speed up this process and hence approximate the low-rank parameters $B$ and $A$ for portable usage. Specifically, randomized SVD creates a random matrix $\boldsymbol{\Omega} \in \mathbb{R}^{d \times c}$ with Gaussian distribution:

$$\boldsymbol{\Omega} \sim \mathcal{N}(\mathbf{0}, \mathbf{I}) \tag{9}$$

After that, we obtain the main column subspace of $\Delta W$ with $\mathbf{Y} = \Delta W \boldsymbol{\Omega}$ to approximate the feature space of original delta weight. Followed by that, we compute an approximation of orthonormal bases of $\Delta W$: $\mathbf{Q} \in \mathbb{R}^{d' \times c}$ using QR decomposition on $\mathbf{Y}$: $\mathbf{Y} = \mathbf{Q}\mathbf{R}$. Based on the orthonormal basis, we obtain the projection of delta weight $\mathbf{B} \in \mathbb{R}^{c \times d}$ on the low-dimensional representation space of $\mathbf{Q}$:

$$\mathbf{B} = \mathbf{Q}^\top \Delta W \tag{10}$$

Then we compute the standard SVD on the smaller matrix $\mathbf{B}$:

$$\mathbf{B} = \hat{\mathbf{U}}' \boldsymbol{\Sigma}' \mathbf{V}'^\top \tag{11}$$

where $\hat{\mathbf{U}}' \in \mathbb{R}^{c \times c}, \boldsymbol{\Sigma}' \in \mathbb{R}^{c \times d}, \mathbf{V}'^\top \in \mathbb{R}^{d \times d}$. We transform back $\hat{\mathbf{U}}'$ to approximate singular vectors $\mathbf{U}' \in \mathbb{R}^{d' \times c}$ as $\mathbf{U}' = \mathbf{Q}\hat{\mathbf{U}}'$. Based on above quantities, we reconstruct the approximated $B$ and $A$ as:

$$B' = \mathbf{U}' \cdot \mathrm{diag}(\boldsymbol{\Sigma}') \quad A' = \mathbf{V}'^\top[1:c,:] \tag{12}$$

where $\mathrm{diag}(\boldsymbol{\Sigma}') \in \mathbb{R}^{c \times c}$ is a diagonal matrix satisfying $\mathrm{diag}(\boldsymbol{\Sigma}')_{i,i} = \boldsymbol{\Sigma}'_{i,i}, \forall\, 0 < i \leq c$ and $\mathrm{diag}(\boldsymbol{\Sigma}')_{i,j} = 0, \forall\, i \neq j$. To determine an appropriate approximation factor $c$, we innovatively introduce **Sim-Search** which searches $c$ through the affinity of remaining components.

## 3.2 SIM-SEARCH

Previous works (Hu et al., 2022; Wang et al., 2024) empirically demonstrate that the subspace similarity between the delta weight $\Delta W$ and the pretrained weight $W$ correlates positively to downstream performances. Meanwhile, Gekhman et al. (2024) also points out that during fine-tuning, LLMs memorize new knowledge by hallucinating itself, which degrades the effectiveness of fine-tuning. Therefore, in NORM, the remaining components should satisfy both two rules:

1. *The remaining $c$ factors should contribute most positively compared to others $c' \neq c$. In other words, these $c$ components contain the least noise.*
2. *The subspace spanned by the remaining $c$ singular vectors should possess the highest subspace similarity with that spanned by the pretrained weight.*

Based on the two rules, we introduce a search step $\tau$ and search the $c = r \cdot \beta$ values ranging from a given start value $s$ to $r$: $\{\tau \cdot s, (\tau + 1) \cdot s, \cdots, r\}$ and perform Equation 9-12 to obtain $B'_c$ and $A'_c$ under different $c$ values. Followed by these newly obtained $B'_c$ and $A'_c$, we reconstruct the delta weight $\Delta W_c$ by $\Delta W_c = B'_c A'_c$ and compute the major $r$ singular vectors again by the random SVD:

$$\mathbf{U}_c, \boldsymbol{\Sigma}_c, \mathbf{V}_c^\top = \mathrm{Random} - \mathrm{SVD}(\Delta W_c) \tag{13}$$

We also use Equation 13 to decompose the pretrained weight $W$ to obtain $\mathbf{U}, \boldsymbol{\Sigma}$ and $\mathbf{V}^\top$. We extract the $r$ left singular vectors of $\mathbf{U}_c$ and $\mathbf{U}$ and compute the subspace similarity as such:

$$\phi_c = \frac{\|\mathbf{U}_{cr}^\top \cdot \mathbf{U}_r\|_F^2}{r} \tag{14}$$

where $\mathbf{U}_{cr} = \mathbf{U}_c[:,:r] \in \mathbb{R}^{d \times r}, \mathbf{U}_r = \mathbf{U}[:,:r] \in \mathbb{R}^{d \times r}$. Based on the computed Grassmann distance $\phi_c$, we select the $c$ value and corresponding reduced delta weights $B'_c$ and $A'_c$ as such:

$$\{c, B'_c, A'_c\} = \arg\max_c \phi_c \tag{15}$$

Finally, these parameters can finally merge back into the pretrained weight to introduce no inference latency: $W' = W + B'_c A'_c, \forall\, W \in \{W\}^p$.

## 4 EXPERIMENTS

In this section, we comprehensively evaluate the proposed NORM method on various downstream domains, including general language understanding, mathematical reasoning, and code generation.

| Model | Method | BBH | MMLU | TydiQA | CQA | TruthfulQA | GSM8K | Logiqa-EN | Average |
|---|---|---|---|---|---|---|---|---|---|
| Qwen2-7B | Base | 41.73 | 66.84 | 45.36 | 74.20 | 57.65 | 87.19 | 40.40 | 59.05 |
| | LoRA | 44.31 | 67.95 | 48.45 | 75.84 | 50.80 | 83.02 | 43.93 | 59.19 |
| | LoRA+ | 42.64 | 67.81 | 52.15 | 78.71 | 50.18 | 83.24 | 43.78 | 59.79 |
| | DoRA | 43.11 | 67.94 | 44.64 | 75.43 | 53.98 | 82.94 | 44.85 | 58.99 |
| | MoRA | 36.31 | 62.43 | 46.70 | 72.65 | 53.37 | 75.44 | 44.09 | 55.85 |
| | TAIA | 44.79 | 66.36 | 46.81 | 75.10 | 58.51 | 85.60 | 45.78 | 60.42 |
| | NoRM | 45.19 | 68.89 | 50.80 | 78.05 | 57.77 | 85.06 | 46.08 | **61.69** |
| Llama3-8B | Base | 40.68 | 59.18 | 40.90 | 71.83 | 63.28 | 77.79 | 41.17 | 56.40 |
| | LoRA | 36.38 | 63.42 | 43.56 | 77.64 | 48.10 | 72.02 | 38.40 | 54.22 |
| | LoRA+ | 40.30 | 63.30 | 38.01 | 75.35 | 37.70 | 73.62 | 36.71 | 52.14 |
| | DoRA | 36.95 | 63.89 | 42.54 | 77.97 | 42.59 | 71.27 | 37.94 | 53.31 |
| | MoRA | – | 25.09 | – | 20.48 | 20.07 | – | 26.42 | 23.01 |
| | TAIA | 35.83 | 62.02 | 47.26 | 76.66 | 54.22 | 77.10 | 37.94 | 55.86 |
| | NoRM | 43.11 | 64.61 | 46.21 | 77.72 | 54.10 | 77.71 | 43.47 | **58.13** |
| Mistral-7B | Base | 39.26 | 54.07 | 30.04 | 66.83 | 56.30 | 55.27 | 36.87 | 48.38 |
| | LoRA | 34.79 | 53.68 | 41.88 | 73.05 | 53.37 | 41.47 | 33.79 | 47.43 |
| | LoRA+ | 32.51 | 56.67 | 39.99 | 70.52 | 42.23 | 45.49 | 38.40 | 46.54 |
| | DoRA | 34.66 | 53.31 | 32.01 | 68.88 | 38.19 | 42.99 | 36.25 | 43.76 |
| | MoRA | 26.69 | 27.58 | 27.40 | 49.14 | 33.66 | 20.24 | 33.18 | 31.13 |
| | TAIA | 38.06 | 54.97 | 39.20 | 71.91 | 53.24 | 50.87 | 38.25 | 49.50 |
| | NoRM | 39.29 | 58.05 | 40.56 | 73.55 | 54.47 | 52.01 | 37.63 | **50.79** |

Table 1: Experiment results on general instruction tuning with Qwen2-7B, Llama3-8B, and Mistral-7B pre-trained models. "–" means a zero performance on specific datasets. "CQA" indicates the CommonsenseQA dataset. All experiments are conducted based on open-sourced codebases. **Bold** represents the best result. The NoRM setting achieves the best results in most datasets.

## 4.1 Experiment Setups

**Training and Evaluation** We choose Llama3-8B-Instruct, Qwen2-7B-Instruct, and Mistral-7B-v0.3-Instruct[2] as the base model. For the decoding strategy, we adopt the zero-shot setting with $temperature = 0$ for reproducible generation. We choose a 100K subset of TÜLU V2 as the general instruction tuning dataset and evaluate each fine-tuning method across various tasks, including symbolic reasoning, commonsense reasoning, knowledge understanding and multi-lingual understanding. Apart from general tuning, we also choose math reasoning and code generation as specific fine-tuning tasks and utilize LLama3-8B as the pretrained model. Specifically, we employ MetaMathQA (Yu et al., 2024b) to fine-tune the base model for math reasoning, which consists of 395K training samples evolved from GSM-8K (Cobbe et al., 2021) and MATH (Hendrycks et al., 2021b). The evaluation sets are the corresponding test sets of GSM-8K and MATH to test models' solving capabilities for math word problems. For the code generation, we utilize `Magicoder-Evol-Instruct-110K` (Wei et al., 2024) as the training data. All fine-tuned models are assessed on HumanEval (Chen et al., 2021) and MBPP (Austin et al., 2021) benchmarks, which contain 164 and 378 high-quality Python text-to-code problems, respectively. For more rigorous evaluation for programming-oriented models, we also test models on HumanEval+ and MBPP+ of the EvalPlus (Liu et al., 2024) benchmark.

**Implementation Details** We choose LoRA (Hu et al., 2022), DoRA (yang Liu et al., 2024), LoRA+ (Hayou et al., 2024), MoRA (Jiang et al., 2024b) as the compared PEFT baselines and TAIA (Jiang et al., 2024a) as the PREFT baseline. All experiments are conducted on 4 NVIDIA A100 GPUs. We use BFloat16 precision and fine-tune all training corpus for 1 epoch. The learning rate is set to 2e-4 and the LoRA rank is set to 64. We use a linear warmup strategy with a 0.03 warmup ratio and a cosine learning rate scheduler. For NoRM's setting, the search step $\tau$ is set to 0.1 and the search range starts at 1. More details can be found in Appendix D.

## 4.2 Main Results

Table 1 and 2 present a comprehensive comparison between various PEFT methods (LoRA, LoRA+, MoRA, and DoRA), and PREFT methods (TAIA and our proposed NoRM) across different training

---

[2]We choose instruction-tuned models instead of base models for higher zero-shot compatibility and more accurate evaluation

| Model | Math | | | Code | | | | Avg. |
|---|---|---|---|---|---|---|---|---|
| | GSM8k | MATH | HumanEval | HumanEval+ | MBPP | MBPP+ | | |
| Pretrained | 77.79 | 31.06 | 53.00 | 48.80 | **71.70** | 60.60 | | 57.16 |
| *PEFT Method* | | | | | | | | |
| Full | 77.86 | 27.86 | 58.50 | 53.70 | 65.30 | 55.00 | | 56.37 |
| LoRA (Hu et al., 2022) | 80.21 | 29.27 | 48.80 | 44.50 | 67.20 | 57.40 | | 54.56 |
| LoRA+ (Hayou et al., 2024) | 78.77 | 28.22 | 56.70 | 52.40 | 68.50 | 58.50 | | 57.18 |
| DoRA (yang Liu et al., 2024) | 80.82 | 29.84 | 51.80 | 46.30 | 66.90 | 57.40 | | 55.51 |
| MoRA (Jiang et al., 2024b) | 63.15 | 19.48 | 43.30 | 39.00 | 47.90 | 39.20 | | 42.01 |
| PREFT *Method* | | | | | | | | |
| TAIA | 79.08 | 32.60 | 59.10 | 53.00 | 69.30 | 60.60 | | 58.95 |
| NORM | **82.79** | **33.76** | **63.40** | **59.80** | 71.40 | **60.80** | | **61.99** |

Table 2: Experimental results on math reasoning and code generation with the Llama3-8b base model. For LoRA+, DoRA and MoRA, we implement them using their open-sourced codebases. **Bold** text represents the best result. The NORM setting achieves the best results in most datasets.

| Model/Method | General | | | Math | | Code | | Avg. |
|---|---|---|---|---|---|---|---|---|
| | BBH | MMLU | TydiQA | GSM8k | MATH | HumanEval (+) | MBPP (+) | |
| Pre-trained | 40.68 | 59.18 | 40.9 | 77.79 | 31.06 | 53.00 (48.80) | 71.70 (60.60) | 53.75 |
| NORM | 43.11 | **64.61** | **46.21** | **82.56** | **34.02** | **63.40 (59.80)** | 71.40 (60.80) | **58.22** |
| *w/* min | 43.14 | 64.36 | 44.54 | 80.36 | 32.58 | 61.60 (54.90) | 69.60 (59.50) | 56.73 |
| *w/* minor | 42.44 | 64.56 | 44.98 | 81.96 | 34.00 | 61.60 (57.30) | 72.00 (**61.10**) | 57.77 |
| *w/* $L^2$ | 43.86 | 63.86 | 44.47 | 79.76 | 31.58 | 57.90 (53.00) | 69.80 (60.60) | 56.09 |
| *w/* cos | **43.91** | 64.09 | 45.93 | 80.06 | 32.48 | 61.60 (56.70) | 70.40 (59.30) | 57.16 |
| *w/* PCA | 29.28 | 63.85 | 40.11 | 79.23 | 29.28 | 54.90 (51.20) | **72.50** (60.10) | 53.38 |

Table 3: Ablation experiments on the selection of major components of NORM. The major components selected by NORM can reserve most LoRA representation but discard most noise.

datasets and evaluation benchmarks. In general instruction tuning, NORM generally performs best across various pre-trained models. Notably, most PEFT-based methods experience parameter redundancy, leading to either minimal gains or even performance degradation. In contrast, PREFT-based methods exhibit superior data efficiency, consistently improving upon base models. Specifically, NORM surpasses LoRA by 4.37 points on MMLU and 3.69 points on LogiQA-EN when applied to Mistral-7B. In downstream tasks such as math reasoning and code generation, NORM also achieves the highest performance among all baselines, delivering a +5.31 improvement over LoRA and a +2.73 gain compared to TAIA. Additionally, PEFT-based methods, including DoRA, LoRA+, and MoRA, fail to surpass vanilla LoRA in instruction-tuning tasks and fall significantly behind the PREFT method, TAIA. In contrast, NORM significantly surpasses TAIA in symbolic reasoning tasks, such as math reasoning and code generation, highlighting its effectiveness in reasoning-intensive scenarios.

## 4.3 ABLATION STUDY

In this section, we investigate the impact brought by the choice of shearing methods and highlight the superiority of the random SVD in NORM and the smart determination of $\beta$ of ***Sim-Search***. We choose five comparatives: (1) *w/* min: NORM but choose the least similar major components; (2) *w/* minor: NORM but choose the minor $c$ components; (3) *w/* $L^2$: NORM with the $L^2$ distance metric; (4) *w/* cos: NORM with the cosine similarity metric; and (5) *w/* PCA: NORM with PCA method for the selection of major components. We use the same settings of §4.1 except subsampling three general evaluation sets covering diverse tasks (BBH, MMLU, and TydiQA) to conduct the ablation experiments. Results in Table 3 demonstrate that random SVD selection outperforms PCA selection. Furthermore, we show that subspace similarity metrics outperform other similarity-based methods, underscoring their precision in measuring the affinity between the retained components and the pre-trained weights. By leveraging the ***Sim-Search*** method, NORM effectively preserves the most relevant components for downstream tasks while maintaining alignment with the base models from a subspace perspective, thereby minimizing the risk of hallucination during fine-tuning.

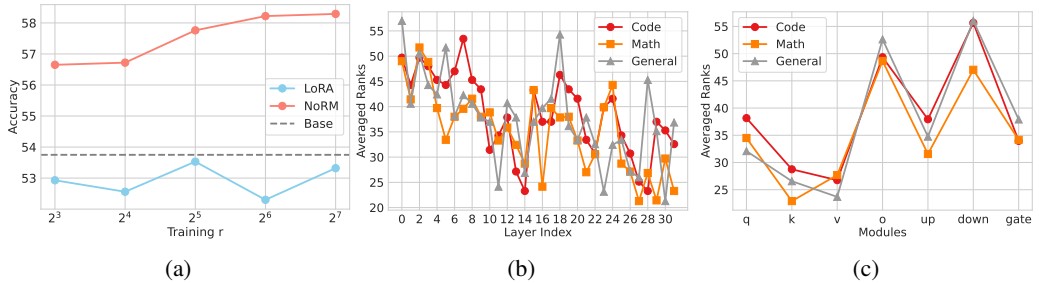

Figure 4: (a) NORM benefits from larger ranks, while vanilla LoRA often obtains lower performance on larger ranks; parameter rank distribution of NORM among layers (b) and modules (c).

## 5 ANALYSIS

In this section, we aim to answer the following research questions (RQ):

**RQ1:** How is the parameter redundancies change with the number of delta parameters?
**RQ2:** What do the sheared parameters distribute over the LLM?
**RQ3:** What is the secret for NORM's high performance over PEFT methods?
**RQ4:** Does NORM scale to other sizes of training data?
**RQ5:** What has been reduced by NORM?

**Response to RQ1: NORM benefits from larger ranks, with a different manner with LoRA.**
To validate the hypothesis that NORM can leverage more tunable parameters and hence gain more improvements, we experiment NORM with an increasing rank sequence: $[8, 16, 32, 64, 128]$ and use the same hyperparameter settings and evaluation datasets described in §4.3. The results, shown in Figure 4a, indicate that compared to LoRA, which usually achieves its highest performance with a middle rank, NORM achieves higher performance with more tunable parameters, which conforms to the general relationship between parameter amounts and performance upgrades. Such distinction also instantiates that vanilla LoRA tuning introduces noise and redundancies into parameters, while NORM can intelligently remove such distraction and largely benefit from the fine-tuning corpus. We also notice flattening improvements when scaling the LoRA ranks to 128 due to an unstable training process; therefore we choose $r = 64$ in our main experiments. Full results are presented in Table 6.

**Response to RQ2: Reduced parameters distribute as §2.2 suggests in most cases.** To visualize the distribution of sheared parameters across the transformer architecture, we compute the reduced rank for each checkpoint in terms of both layers and modules. Results in Figure 4b demonstrate that in upper layers, NORM tends to remain fewer parameters, which is accordant with the conclusion in §2.2. In contrast, in the half layers (around 16-18), NORM behaves to maintain large ranks. Such distinct manner derives from the remaining strategy of §2.2 (random drop) and NORM (random SVD). Besides, NORM remains as much down_proj and o_proj ranks as Figure 2c indicates in a module-wise perspective. NORM maintains ∼ 50% parameters for self-attention modules, which is the configuration obtaining the highest performance in §2.2. In conclusion, NORM intelligently erases the redundancy following its distribution in LLMs and achieves superior performance.

**Response to RQ3: NORM forgets less and learns more.** We unveil the secret of NORM from the forgetting perspective, where we benchmark the vanilla LoRA method and NORM on memorizing pre-trained knowledge. We follow Kalajdzievski (2024) to use WikiText-103 test dataset (Merity et al., 2016) as the evaluation set since it has already served as the pre-training corpus for most LLMs. We use the cross-entropy loss as the metric to test the base model, LoRA-tuned model and NORM-tuned counterpart taking the Llama3-8B-Instruct as the backbone. Results in Table 4 demonstrate that NORM outperforms LoRA and the base model with a large margin, no matter what training data it leverages. Notably, LoRA-tuned models are generally inferior to the base model, indicating that LoRA tuning still results in forgetting problems, albeit relatively few tunable parameters. In contrast, NORM discards the hallucinatory contents accompanied by the learning of new knowledge and hence strengthens the memorization of internal knowledge.

| Method | Training Data | | |
|---|---|---|---|
| | General | Coding | Math |
| Base | | 3.7016 | |
| LoRA | 3.7102 | 3.7512 | 3.7323 |
| NORM | **3.5992** | **3.6450** | **3.6519** |

Table 4: Forgetting loss on WikiText-103 test dataset. NORM reduces hallucination and hence gains superior memorization of internal knowledge over other baselines.

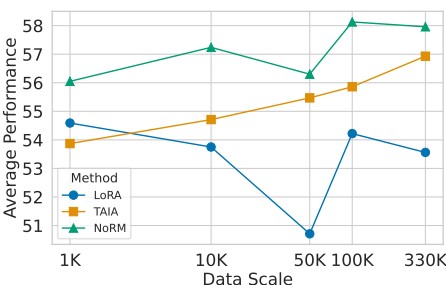

Figure 5: Performance of fine-tuned Llama3-8b using different numbers of TÜLU V2 training samples.

**Response to RQ4: NORM demonstrates superiority across various sizes of fine-tuning datasets.** Considering that in practical conditions, access to extensive fine-tuning datasets is frequently limited, we compare NORM to LoRA and TAIA for fine-tuning LLaMA3-8B with a range of instruction-tuning sample sizes, specifically [1K, 10K, 50K, 100K, 330K], with 330K being the full size of TÜLU V2. We visualize the average performance of each method in Figure 5 and present the full results in Table 7. The results show that NORM consistently outperforms LoRA and TAIA across all training sample sizes. With 10K training samples, NORM surpasses LoRA and TAIA by margins of 3.49 and 2.53, respectively. Even when the training size is reduced to 1K, NORM maintains its lead with advantages of 1.46 and 2.18 over LoRA and TAIA, respectively. This demonstrates that our methods persistently enhance performance over LoRA and TAIA, regardless of the training sample volume.

**Response to RQ5: NORM reduces the amplification ratio of already-emphasized directions of pretrained weights.** In this research question, we investigate the relationship between $W$ and $\Delta W$. We answer this question by projecting $W$ onto the $r$-dimensional subspace of $\Delta W$ by computing $U^\top W V$ with $U/V$ being the left/right singular-vector matrices of $\Delta W$. We compare the Frobenius norm between $\|\Delta W\|$ and $\|U^\top W V\|$ and compute the amplification factor (AF) as $\frac{\|\Delta W\|}{\|U^\top W V\|}$. To demonstrate that NORM further inhibits the already-amplified directions of $W$ to be activated, we also compute the reverse amplification factor (RAF) by projecting $W$ onto the last $d - r$-dimensional subspace: $\frac{\|\Delta W\|}{\|U_{d-r}^\top W V_{d-r}\|}$. We follow the setting of Hu et al. (2022) and draw two main conclusions from Table 5. First, both methods amplify main features that are already in $W$ with negligible distinction. Second, NORM reduces substantially the amplification factor of directions already emphasized in $W$. These two findings suggest that NORM potentially maintains most contributing features of LoRA parameters but further suppresses the amplification of already-emphasized features.

## 6 CONCLUSION

In this paper, we first use sufficient empirical experiments to reveal the general parameter redundancies among LoRA parameters, especially among model layers and specific modules. Built on these insights, we set up PREFT, a novel tuning framework that highlights the utilization of LoRA parameters by removing intrinsic redundancies without sacrificing training and inference efficiency. Under this framework, we propose NORM to reserve the most contributing components of LoRA parameters which possess the highest subspace similarity with pre-trained weights, with a novel ***Sim-Search*** method. Experiment results show that NORM achieves superior improvements on various domains, verifying its application in diverse domains by enhancing the capacity of high LoRA ranks.

ACKNOWLEDGMENTS

We thank the anonymous reviewers for their insightful comments and suggestions. This work is supported by National Key R&D Program of China (No. 2022ZD0162101), National Natural Science Foundation of China (No. 62106140) and STCSM (No. 21511101100, No. 22DZ2229005).

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

| | LoRA | | | | NoRM | | | |
|---|---|---|---|---|---|---|---|---|
| | $\Delta W_q$ | $W_q$ | AF↑ | RAF↓ | $\Delta W_q$ | $W_q$ | AF↑ | RAF↓ |
| $\|U^\top W_q V\| = $ $\|W_q\| = 74.0$ | 1.16 | 34.26 $\|\Delta W_q\| = 2.1875$ | 1.88 | 0.03 | 0.33 | 21.99 $\|\Delta W_q\| = 0.6133$ | 1.83 | 0.008 |

Table 5: The Frobenius norm of $U^\top W_q V$ where $U$ and $V$ are the left/right top $r$ singular vector directions of either $\Delta W_q$ and $W_q$. The weight matrices are taken from the 16th layer of Llama3-8B. "AF" and "RAF" indicate the amplification factor and reverse amplification factor, respectively.

| Param Config | Method | General | | | Math | | Code | | Avg. |
|---|---|---|---|---|---|---|---|---|---|
| | | BBH | MMLU | TydiQA | GSM8k | MATH | HumanEval (+) | MBPP (+) | |
| – | Base | 40.68 | 59.18 | 40.90 | 77.79 | 31.06 | 53.00 (48.80) | 71.70 (60.60) | 53.75 |
| r=8,α=16 | LoRA | 34.43 | 63.89 | 43.72 | 80.82 | 29.06 | 53.00 (48.20) | 66.90 (56.30) | 52.93 |
| | NoRM | 41.31 | 64.43 | 45.77 | 81.80 | 32.12 | 59.10 (54.90) | 70.90 (59.50) | 56.65 |
| r=16,α=32 | LoRA | 34.88 | 61.45 | 43.32 | 81.20 | 28.66 | 50.60 (45.70) | 68.50 (58.70) | 52.56 |
| | NoRM | 39.79 | 63.70 | **46.43** | 81.58 | 32.74 | 61.00 (56.70) | 69.80 (58.70) | 56.72 |
| r=32,α=64 | LoRA | 36.52 | 61.03 | 46.80 | 79.00 | 29.90 | 52.40 (48.80) | 68.80 (58.50) | 53.53 |
| | NoRM | 41.64 | 62.44 | 46.28 | 82.56 | 32.84 | 62.80 (58.50) | 72.20 (60.60) | 57.76 |
| r=64,α=128 | LoRA | 36.38 | 63.42 | 43.56 | 80.21 | 29.27 | 48.80 (44.50) | 67.20 (57.40) | 52.30 |
| | NoRM | **43.11** | 64.61 | 46.21 | 82.56 | **34.02** | **64.00 (59.80)** | 70.40 (59.30) | 58.22 |
| r=128,α=256 | LoRA | 36.18 | 63.77 | 46.99 | 80.74 | 30.48 | 48.80 (45.10) | 68.80 (59.00) | 53.32 |
| | NoRM | 42.30 | **64.86** | 45.30 | **83.17** | 32.92 | 63.40 (58.50) | **72.80 (61.40)** | **58.29** |

Table 6: Full results on the analysis on tunable parameters. The tunable parameters are increased incrementally to validate NoRM's behavior and effectiveness.

# A  RELATED WORK

**Parameter efficient fine-tuning**  Full fine-tuning effectively adapts large language models to downstream tasks (Fan et al., 2025) but requires substantial computational resources as model size and task numbers increase. To mitigate this, Parameter-Efficient Fine-Tuning (PEFT) methods have been introduced. These methods freeze the base language models and modify only a minimal number of parameters during training, achieving comparable or even superior performance with limited fine-tuning data. Among these methods, Adapter-Tuning (Rebuffi et al., 2017; Houlsby et al., 2019; Lin et al., 2020; Pfeiffer et al., 2021), Prefix-tuning (Li & Liang, 2021), Prompt-Tuning (Lester et al., 2021), P-Tuning (Liu et al., 2023b) and P-Tuning-v2 (Liu et al., 2022) were proposed to reduce fine-tuning costs before the era of LLMs. However, these methods introduce additional priors and significant inference latency. In contrast, Low-Rank Adaptation (LoRA)(Hu et al., 2022) and its variant, Weight-Decomposed Low-Rank Adaptation (DoRA)(yang Liu et al., 2024), take a different approach. LoRA updates original parameters with two low-rank matrices without assuming any specific task or architecture, eliminating inference latency by merging back these two matrices to the original weight. DoRA extends this by incorporating weight decomposition into magnitude and direction, achieving performance comparable to full fine-tuning. LoRA+ (Hayou et al., 2024) proposed to adopt adaptive update strategies for each low-rank parameter. MoRA (Jiang et al., 2024b) proposed to incorporate square high-rank tunable parameters to achieve both efficiency and high rank. Nonetheless, most LoRA-like methods require complex hyperparameter settings, which hinders their generalization.

**Parameter redundancies**  Previous works have verified that in pretrained language models, parameters contain sufficient redundancies. Dalvi et al. (2020) show that redundancies exist in layers and neurons and vary among downstream tasks. Bhojanapalli et al. (2021) demonstrate that there exist substantial redundancies among transformer multi-head attention modules. He et al. (2024) empirically verify the redundancies of LLMs within self-attention and multi-layer perceptron (MLP) modules and leverage these redundancies to speed up inference. Men et al. (2024) also leverage such layer redundancies to boost the inference speed yet sacrificing limited performance. However, these works only analyze the redundancies among pretrained LLMs, but less focus on fine-tuned delta parameters, especially in so-called low-rank parameters. Yu et al. (2024a) leverage the delta parame-

ter redundancies to perform model merging without performance degradation. Jiang et al. (2024a) attempt to remove the delta feed-forward low-rank parameters to adapt LLMs to out-of-domain tasks. In this work, we intend to unveil the parameter redundancies among delta low-rank parameters and leverage such redundancies to improve fine-tuned models with more fine-grained practice.

**Limitations and drawbacks of fine-tuning**  Fine-tuning is a common method for adapting large language models (LLMs) to various downstream tasks. However, it comes with significant drawbacks, including hallucination, harmful outputs, catastrophic forgetting, and safety concerns. Gekhman et al. (2024) noted that fine-tuning can lead models to produce factually inaccurate responses, as the training process encourages the generation of information not grounded in the model's pre-existing knowledge. Additionally, supervised fine-tuning for specific tasks often results in catastrophic forgetting of the initial alignment (Luo et al., 2023b) and creates trade-offs between helpfulness and harmlessness (Bai et al., 2022). Kumar et al. (2024) also highlighted that fine-tuning significantly reduces the resistance of LLMs to jailbreaking, thereby increasing their vulnerability. Even when carefully curated fine-tuning datasets are used, Qi et al. (2023) demonstrated that well-aligned LLMs often become less safe and more prone to harmful behavior, with issues exacerbated by red-teaming in the tuning data. In contrast, NORM addresses many of these drawbacks while enhancing helpfulness through fine-tuning. By focusing on retaining the most similar delta components relative to the base weights, NORM offers a robust solution to the challenges associated with traditional fine-tuning approaches.

## B  LIMITATIONS

We notice that NORM gains smaller performance gains with enlarged LoRA ranks. We hypothesize that although NORM removes noisy components of updated LoRA parameters, it still cannot fully separate the redundant parts, which causes the distracting parts to interfere with downstream performances. Besides, we currently only apply NORM to the inference-preprocessing stage. The introduction of NORM to the training stage may support a more convenient application of NORM and further improvements over LoRA and TAIA.

## C  FUTURE WORK

NORM succeeds in discarding noisy components of LoRA parameters by selecting the most contributing parts through *Sim-Search*. To enlarge the application of NORM, the next research direction is to extend NORM to the full fine-tuning scenario. Just as Yu et al. (2024a) indicates, the full fine-tuned parameters also contain extra but useless parameters that can be smartly reduced by appropriate PREFT methods. Besides, an adaptive maintaining strategy instead of the coarse separation can improve the downstream application of NORM, by picking the most appropriate components of specific prompts automatically. We hope our work can provide inspiration on how to improve the parameter utilization of LLMs to bootstrap the performance of LLMs on downstream tasks.

## D  EXPERIMENT DETAILS

### D.1  PEFT SETTINGS

We add a LoRA module for each linear layer except the language model head and embedding layer, resulting in the following target modules: [q_proj, k_proj, v_proj, o_proj, gate_proj, up_proj, down_proj]. The LoRA $\alpha$ of each experiment is set to twice of the LoRA rank suggested by Hu et al. (2022).

### D.2  EVALUATION DETAILS

Our evaluations contain different types of metrics, including Exact Match (EM) and Multiple Choice Accuracy (Acc). For EM metric, we extract the contents followed by The answer is to reduce evaluation biases. For different datasets, we adopt different evaluation prompts after the original problem description:

1. **MATH** (Hendrycks et al., 2021c), **GSM-8K** (Cobbe et al., 2021), **BIG-Bench Hard** (Suzgun et al., 2022) (BBH), **SVAMP** (Patel et al., 2021): `Please format the final answer at the end of the response as:  The answer is {answer}.`
2. **HumanEval (+)** (Chen et al., 2021), **MBPP (+)** (Austin et al., 2021; Liu et al., 2022), **TruthfulQA** (Lin et al., 2022): None.
3. **MMLU** (Hendrycks et al., 2021a), **LogiQA** (Liu et al., 2020): `Please answer with option letter directly, do not output other information.`
4. **CommonsenseQA** (Talmor et al., 2019): `Let's think step by step.  Please format the final answer at the end of the response as:  The answer is {answer}.`

We use greedy decoding to maintain that all results are reproducible.

### D.3 Test Sets Description

We here describe the used ten evaluation sets:

1. **MATH** (Hendrycks et al., 2021c) is a collection of challenging competition mathematics problems containing 5,000 problems in the test set. Each problem in MATH has a full step-by-step solution which can be used to teach models to generate answer derivations and explanations.
2. **GSM-8K** (Cobbe et al., 2021) is a collection of 1,273 math-reasoning problems with varying difficulty. Each problem requires the model to conduct single or multi-hop reasoning to derive the correct answer.
3. **SVAMP** (Patel et al., 2021) are much simpler datasets compared to MATH, which both test models' math problem-solving ability. It contains 1,221 problems which are all solvable with one or two simple equations.
4. **BIG-Bench Hard** (Suzgun et al., 2022) (BBH) is a collection of 23 challenging tasks from BIG-Bench. The 6,511 problems are the tasks for which prior language model evaluations did not outperform the average human-rater.
5. **CommonsenseQA** (Talmor et al., 2019) is to test models' ability to answer questions using only the parameterized knowledge instead of the context knowledge. It contains 1,000 problems sourced from ConceptNet (Speer et al., 2017).
6. **LogiQA** (Liu et al., 2020) collects questions about natural language inference (NLI) and requires models to infer the conclusion based on provided premises. It contains 653 problems for both English and Chinese subsets.
7. **TruthfulQA** (Lin et al., 2022) is for testing models' ability to produce truthful answers. The 817 questions that span 38 categories benchmark models' refusal to generate false answers like humans.
8. **MMLU** (Hendrycks et al., 2021a) is to measure LLM's multitask accuracy, which contains 14,421 problems. The test covers 57 tasks including elementary mathematics, US history, computer science, law, and more. To attain high accuracy on this test, models must possess extensive world knowledge and problem-solving ability.
9. **HumanEval** (Chen et al., 2021) contains 164 human-checked python-oriented programming problems to evaluate models' code generation ability. The **HumanEval+** (Liu et al., 2024) version creates more comprehensive test cases to produce a more fair evaluation result.
10. **MBPP** (Austin et al., 2021) contains 500 python coding problems, where each problem requires the model to generate correct python functions. The **MBPP+** (Liu et al., 2024) version creates more comprehensive test cases to produce a more fair evaluation result.

## E Further Experiments

### E.1 Comparison with More PEFT Methods

In this section, we add more PEFT baselines to further validate the effectiveness of NoRM. We select LoRA-pro (Wang et al., 2025), which modifies LoRA's training mechanism, and two methods adopting the philosophy of SVD. One method, PiSSA (Meng et al., 2024), fixes the minor components

| Data Size | METHOD | BBH | MMLU | TydiQA | CQA | TruthfulQA | GSM-8K | LogiQA en | Average |
|---|---|---|---|---|---|---|---|---|---|
| – | Base | 40.68 | 59.18 | 40.90 | 71.83 | 63.28 | 77.79 | 41.17 | 56.40 |
| 1K | LoRA | 34.88 | 63.35 | 49.18 | 76.41 | 45.78 | 74.75 | 37.79 | 54.59 |
| | TAIA | 19.92 | 62.18 | 46.12 | 76.17 | 53.00 | 77.33 | 42.40 | 53.87 |
| | NORM | 36.12 | 62.11 | 45.77 | 76.74 | 52.02 | 76.12 | 43.47 | 56.05 |
| 10K | LoRA | 36.03 | 64.72 | 43.14 | 75.18 | 49.69 | 70.89 | 36.56 | 53.75 |
| | TAIA | 26.62 | 64.41 | 46.46 | 76.66 | 52.14 | 77.33 | 39.32 | 54.71 |
| | NORM | 39.69 | 64.85 | 48.63 | 76.25 | 52.75 | 75.97 | 42.55 | 57.24 |
| 50K | LoRA | 33.67 | 57.29 | 40.42 | 74.86 | 45.90 | 69.83 | 33.03 | 50.71 |
| | TAIA | 36.63 | 61.69 | 44.79 | 76.41 | 52.75 | 76.19 | 39.78 | 55.47 |
| | NORM | 40.06 | 62.97 | 43.77 | 76.49 | 52.14 | 76.42 | 42.24 | 56.30 |
| 100K | LoRA | 36.38 | 63.42 | 43.56 | 77.64 | 48.10 | 72.02 | 38.40 | 54.22 |
| | TAIA | 35.83 | 62.02 | 47.26 | 76.66 | 54.22 | 77.10 | 37.94 | 55.86 |
| | NORM | 43.11 | 64.61 | 46.21 | 77.72 | 54.10 | 77.71 | 43.47 | 58.13 |
| 330K | LoRA | 37.01 | 60.70 | 45.09 | 74.28 | 50.80 | 67.85 | 39.17 | 53.56 |
| | TAIA | 41.68 | 62.63 | 45.69 | 74.53 | 57.53 | 76.65 | 39.78 | 56.93 |
| | NORM | 41.42 | 63.47 | 45.16 | 76.17 | 58.26 | 78.09 | 43.16 | 57.96 |

Table 7: Full experiment results of data scaling of NORM. NORM consistently maintains the leading performance across all data sizes of TÜLU V2.

| Method | BBH | MMLU | TydiQA | CommonsenseQA | TruthfulQA | GSM8K | Logiqa en | Average |
|---|---|---|---|---|---|---|---|---|
| Base | 40.68 | 59.18 | 40.90 | 71.83 | 63.28 | 77.79 | 41.17 | 56.40 |
| LoRA | 36.38 | 63.42 | 43.56 | 77.64 | 48.10 | 72.02 | 38.40 | 54.22 |
| LoraPro | 37.31 | 61.72 | 45.21 | 74.77 | 47.49 | 63.38 | 33.79 | 51.95 |
| PiSSA | 33.16 | 59.15 | 44.37 | 75.92 | 48.71 | 69.98 | 38.71 | 52.86 |
| AdaLora | 29.80 | 59.27 | 50.59 | 73.30 | 48.71 | 71.87 | 42.40 | 53.71 |
| TAIA | 35.83 | 62.02 | 47.26 | 76.66 | 54.22 | 77.10 | 37.94 | 55.86 |
| NORM | 43.11 | 64.61 | 46.21 | 77.72 | 54.10 | 77.71 | 43.47 | **58.13** |

Table 8: Comparison of NORM with more PEFT baselines, especially SVD-based methods, including AdaLora and PiSSA. NORM performs best among these PEFT methods with Llama3-8B as the base model.

of base weights and uses principle components to initialize LoRA parameters and the other baseline AdaLora (Zhang et al., 2023) adaptively allocates the parameter budget among weight matrices. We select Llama3-8B as the base model and choose TÜLU V2 as the training data. Table 8 depicts the comparison of NORM with these PEFT baselines. These baselines consistently underperform vanilla LoRA when taking an instruction-tuning model as the backbone, while PREFT based NORM show non-negligible improvements. This also indicates that although NORM leverages SVD to operate parameters, SVD tailored at redundancy reduction is more advanced than either the initialization of parameters or update budgets.

## E.2 UTILIZATION IN CONTINUAL LEARNING

As NORM reduces redundant parts of the fine-tuned parameter with the largest subspace overlap with pre-trained parameters, it intrinsically avoids severe knowledge-forgetting problems due to internal distribution shifts. As a result, NORM has significant resistance against catastrophic forgetting dilemma, which leads to performance improvement over vanilla LoRA tuning under continual learning scenarios. We perform experiments following the experimental setup of OLoRA (Wang et al., 2023). Specifically, we adopt the specified training order for continuous learning and evaluate our approach on the StandardCL (Zhang et al., 2015) benchmark. We use LLama3 8B as the base model. The training orders are as follows: (1) Order 1: dbpedia → amazon → yahoo → ag; (2) Order 2: dbpedia → amazon → ag → yahoo; (3) Order 3: yahoo → amazon → ag → dbpedia. This configuration ensures consistency with prior work and allows for a fair comparison of performance across different orderings. We adopt Accuracy as the evaluation metric and show the performance below in Table 9. We observe a substantial improvement over standard LoRA tuning. This enhancement stems from NoRM's ability to preserve the LoRA components most aligned with the pre-trained weights, effectively mitigating the issue of catastrophic forgetting. By prioritizing

| Model | Order1 | Order2 | Order3 | Average |
|-------|--------|--------|--------|---------|
| LoRA  | 65.34  | 74.56  | 70.44  | 70.11   |
| NoRM  | **78.88** | **80.08** | **78.76** | **79.24** |

Table 9: NORM in the continual learning setting.

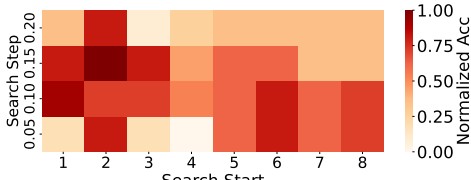

Figure 6: Normalized performance on HumanEval (+) with varying search steps and search ranges. Lower search steps and wider search ranges generally find more similar components and perform outstandingly.

these key components, NoRM ensures better retention of prior knowledge throughout the continuous learning process.

### E.3 LARGE SEARCH RANGES AND FINE-GRAINED SEARCH STEPS BRING FURTHER IMPROVEMENTS.

To rationalize the hyperparameter settings of **Sim-Search**, including the search step $\tau$ and search start $s$, we perform a grid search on these two parameters. We use normalized accuracy described in §2.2 as the evaluation metric and take HumanEval (+) as the evaluation set for simplicity. The search start $s$ is searched in $\{1, 2, \ldots, 8\}$ and $\tau$ is searched in $\{0.05, 0.10, 0.15, 0.20\}$.

Figure 6 presents that with larger search ranges (lower search starts), NORM gains further improvements. Meanwhile, relatively lower search steps generally bring higher results for NORM. Although the setting $s = 2, \tau = 0.15$ brings the best performance on the HumanEval (+) dataset, we still take the setting $s = 1, \tau = 0.1$ for all experiments as this setting is already acceptable enough and we leave the intelligent selection of search parameters for future work.

