# OpenReview forum: "Fine-tuning with Reserved Majority for Noise Reduction"
_ICLR.cc/2025/Conference — ICLR 2025 Spotlight_

### Official Review · Reviewer_FpRJ · 2024-10-28

**Soundness:** 3
**Presentation:** 2
**Contribution:** 2
**Rating:** 6
**Confidence:** 5

**Summary:**

The paper introduces a parameter-efficient fine-tuning framework called Parameter-Redundant Fine-Tuning (PREFT), focusing on reducing noise in LoRA-tuned models through redundancy management. It proposes Noise Reduction with Reserved Majority (NORM), which enhances LoRA by isolating valuable components while removing redundant parts using singular value decomposition. NORM employs Sim-Search to identify the most relevant parameters for fine-tuning by analyzing their similarity with pre-trained weights, reducing hallucinations that degrade model performance.

**Strengths:**

1. While many concurrent papers focus on pushing the limits of parameter efficiency, this paper delves into another interesting aspect of LoRA: eliminating redundancies to prevent hallucinations.
2. Experimental results show that this method yields promising outcomes.

**Weaknesses:**

1. The methodology section is somewhat confusing due to the arbitrary use of notations. For example, in Line 297, does the starting value $s$ have any connection with $s_1, s_2$ mentioned in Line 155? Also, what does $\mathbf{U}_{cr}$ represent in Equation 14?
2. Although the LLM fine-tuning tasks are practical and valuable, the evaluation methods are not very standardized, and therefore the results are not entirely convincing. I recommend that the authors conduct experiments on both the GLUE[1] and VTAB[2] benchmarks to further validate the effectiveness of the proposed method.

[1] GLUE. https://gluebenchmark.com/

[2] Visual Task Adaptation Benchmark. https://google-research.github.io/task_adaptation/

3. Having conducted experiments in this area, I have found that the evaluation methods used for fine-tuned LLMs are often unconvincing （e.g., some evaluations rely on asking ChatGPT for a rating). Therefore, I strongly recommend that the authors perform experiments on established benchmarks like GLUE and VTAB to provide more robust evaluations. However, as anticipated, their method did not excel on GLUE. Additionally, the authors offer only intuitive reasons for not conducting experiments on VTAB, which raises doubts about the actual effectiveness of their proposed method. Consequently, I cannot support acceptance of this work in its current form. (**Addressed**)

**Questions:**

Please refer to the weakness part.

---

> ### Author Response · Authors · 2024-11-17
> **Response to reviewer**
>
> Thank you for your thoughtful review and insightful questions! We are pleased to hear that you find our proposed NoRM both novel and useful, as well as our PREFT fine-tuning framework engaging. Below, we provide detailed responses to the questions and concerns raised in your review.
>
> > L297 Notation confusion
>
> **Our response** : Here $s$ is the search start of Sim-Search and it is irrelevant to the L155 notation. We apologize for the confused notation in L155 and we changed notations in L155 to more suitable ones in the revised manuscript.
>
> > Equation 14 confusion
>
> **Our response** : We apologize for the confusion. In this context, $\mathbf{U}_{cr}=\mathbf{U}_c[:,:r]$ represents the left $r$ singular vectors of $\mathbf{U}_c$. To enhance clarity, we have updated this notation to a more appropriate form in Line 307 of the revised submission.
>
> > Performance of NoRM in GLUE and VTAB
>
> **Our response**: Thanks for your suggestion. We would like to clarify that NoRM is specifically designed for the fine-tuning of large language models and is not applicable to vision tasks. As a result, we are unable to validate the effectiveness of NoRM on VTAB. We appreciate your understanding.
> Below, we present the performance of NoRM alongside other methods on the GLUE benchmark. For evaluation, we selected six subsets: MNLI, MRPC, SST-2, CoLA, RTE, and QQP.
>
> | Model | CoLA  | MNLI  | RTE   | MRPC  | SST-2 | QQP   | Avg    |
> |-------|-------|-------|-------|-------|-------|-------|--------|
> | LoRA  | 78.62 | 92.06 | 84.48 | 68.38 | 96.56 | 87.57 | 84.61  |
> | TAIA  | 80.54 | 91.18 | 83.03 | 33.09 | 96.22 | 88.05 | 78.68  |
> | NoRM  | 82.84 | 91.89 | 83.75 | 68.38 | 95.64 | 89.43 | **85.32**  |
>
> The results show that NoRM outperforms both LoRA and the PREFT-based method TAIA on the selected subsets, further validating the effectiveness of NoRM.

---

> > ### Comment · Reviewer_FpRJ · 2024-11-20
> >
> > Thanks for the rebuttal.
> >
> > I wonder why NoRM could not be applied to vision tasks? Could you explain more?

---

> > > ### Author Response · Authors · 2024-11-20
> > > **Response to reviewer**
> > >
> > > Thank you for your reply.
> > >
> > > We understand that VTAB serves as a benchmark for evaluating visual representation models. Our method, NoRM, is specifically designed for generative models, including large language models (e.g., Llama) and large vision-language models (e.g., Llava), rather than representation learning models. NoRM operates by comparing a fine-tuned model with its pre-trained counterpart to identify and retain the most relevant LoRA parameters. This process is closely tied to the generative characteristics and architecture of these models, making it less suited to tasks centered on visual representation.
> > >
> > > In the case of vision representation tasks, the focus is on learning efficient and generalizable feature embeddings. To adapt NoRM to these tasks would require significant changes to account for their differing objectives and architectures. The method's dependency on generative modeling dynamics makes it incompatible with the evaluation criteria and training frameworks typically used in representation learning.
> > >
> > > For vision generative models such as Llava or Qwen-VL, we believe NoRM has strong potential for application. However, due to time and computational resource constraints, we have not yet implemented a comprehensive multi-modal evaluation framework to benchmark NoRM against other fine-tuning techniques in this domain. We are eager to pursue this direction in future work and look forward to demonstrating its utility in multi-modal settings.
> > >
> > > Thank you again for your thoughtful feedback.

---

> > > > ### Comment · Reviewer_FpRJ · 2024-11-20
> > > >
> > > > Thank you for the explanation.
> > > >
> > > > However, I'm still not entirely convinced. Technically, if we set aside the semantic meanings, ViT processes an image as a sequence, similar to how a sentence is handled in NLP. So, what is technically or architecturally preventing the application of NoRM in this context? If your argument is that the distinction lies between generative and discriminative models, why is NoRM applicable to GLUE tasks, which are also discriminative in nature?

---

> > > > > ### Author Response · Authors · 2024-11-20
> > > > > **Response to reviewer**
> > > > >
> > > > > Thank you for your thoughtful feedback, and we apologize for any confusion in our previous response. Our paper focuses on unveiling the redundancy during fine-tuning of generative language models (LLMs) and proposing  NoRM to mitigate it. The primary interest and focus in this work are especially on generative LLMs.
> > > > >
> > > > > While it is true that Vision Transformers (ViT) process images as sequences similar to how sentences are handled in NLP, there are fundamental differences between visual and textual data that affect how methods like NoRM can be applied. Adapting NoRM to vision tasks would require significant adjustments to account for the different nature of visual data and the architectural nuances of vision models. Visual data involves spatial hierarchies and local feature correlations that differ from the sequential dependencies in textual data, necessitating different strategies for redundancy reduction.
> > > > >
> > > > > Regarding the applicability of NoRM to discriminative tasks like GLUE, we want to clarify that although GLUE is naturally a discriminative task, we use generative LLMs to test GLUE in our experiments. Specifically, we employ a technique involving language logit bias, where we constrain the model to generate just one token by adding a high logit bias to the answer tokens "A", "B", "C", or "D". This approach enforces the LLM to generate a single token from the provided set, and we then compare the generated token with the ground truth to evaluate performance. This method is a standard and popular way of handling multiple-choice questions with generative language models. By doing so, we utilize generative LLMs in a manner that aligns with our focus on generative models while assessing their capabilities on tasks that are traditionally discriminative.
> > > > >
> > > > > We acknowledge that vision tasks are important and agree that extending NoRM to vision tasks is both interesting and valuable. In fact, we are currently conducting adjustments to adapt NoRM to vision models like ViT. However, this adaptation involves significant changes to address the unique characteristics of visual data and is beyond the scope of our current paper.

---

> ### Author Response · Authors · 2024-11-23
> **Response to reviewer**
>
> We have completed the adaptation of NoRM in VTAB.
>
> | Method | cifar | caltech101 | dtd  | oxford_flowers102 | oxford_iiit_pet | svhn | sun397 | patch_camelyon | eurosat | resisc45 | diabetic_retinopathy | clevr_count | clevr_dist | dmlab | kitti | dsprites_loc | dsprites_ori | smallnorb_azi | smallnorb_ele | Average |
> |--------|-------|------------|------|-------------------|-----------------|------|--------|----------------|---------|----------|----------------------|-------------|------------|-------|-------|--------------|--------------|---------------|---------------|---------|
> | LoRA   | 85.9  | 92.2       | 82.2 | 99.7              | 94.5            | 64.1 | 63.6   | 88.8           | 95.0    | 90.5     | 76.6                 | 97.7        | 65.3       | 49.0  | 71.0  | 90.6         | 63.0         | 37.1          | 52.3          | 76.79   |
> | NoRM   | 84.5  | 97.5       | 81.0 | 100.0             | 95.0            | 76.5 | 61.5   | 84.0           | 97.5    | 89.0     | 76.5                 | 93.0        | 64.0       | 56.5  | 70.0  | 92.0         | 57.5         | 36.2          | 50.0          | **76.96**   |
>
> NoRM performs on par with LoRA. As parameter redundancy is observed in the fine-tuning of generative large language models, the method should undertake further adjustments so as to improve more upon LoRA tuning.
> However, the primary objective of this paper is to uncover the parameter redundancy in fine-tuning a generative large language model. Consequently, the potential application of NoRM to vision tasks and other discriminative tasks is considered beyond the scope of this work and is left for future exploration.
>
> Additionally, we have noted that your assessment of our paper was downgraded without clear justification or clarification. We would like to emphasize that this revision appears inconsistent with your initial evaluation, and we respectfully request further elaboration on the rationale behind this change.
> **We hope that you can provide an open, fair, and objective evaluation of the article and rebuttal, with any changes in the scoring being transparent, consistent and logically justified.**

---

> > ### Comment · Reviewer_FpRJ · 2024-11-23
> >
> > I downgraded my score **due to a reasonable concern, as detailed in weakness 3, which has been updated with the score change**. The authors' accusation that I am "not open, fair, objective, transparent, consistent and logically justified" is unacceptable and offensive.
> >
> > Concerning the paper itself, the VTAB results have reinforced my inclination toward rejection. As previously mentioned, the evaluation methods for LLM fine-tuning are not yet standardized. The performance of NoRM on established benchmarks like GLUE and VTAB is not particularly strong, with LoRA and NoRM alternately outperforming each other across different datasets. This inconsistency diminishes the perceived performance advantages of NoRM. Without clear performance or efficiency benefits, the proposed method does not appear sufficiently compelling.
> >
> > Based on these considerations, I will maintain my current score.

---

> > > ### Author Response · Authors · 2024-11-23
> > > **response to reviewer**
> > >
> > > 1. Evaluation Methods and Established Benchmarks
> > >
> > > You raised a concern that evaluation methods for fine-tuned LLMs are often unconvincing, citing examples where evaluations rely on asking ChatGPT for ratings. We fully agree that subjective evaluations can be problematic and may not provide reliable assessments of model performance. However, we want to clarify that our evaluation methodology does not involve subjective assessments from models like ChatGPT. **All the generative tasks presented in our paper are evaluated using objective metrics, specifically accuracy.** Moreover, it is not accurate to say that the tasks we used are "not established." The tasks we employed are standard benchmarks in the field of natural language generation and have been utilized by numerous prior works. These benchmarks are widely recognized and accepted within the research community, ensuring that our results are both credible and comparable to existing literature.
> > >
> > > 2. Performance on GLUE and VTAB Benchmarks
> > >
> > > We appreciate your recommendation to perform experiments on established benchmarks like GLUE and VTAB to provide more robust evaluations. Our method is specifically designed for generative large language models and aims to enhance performance on generative tasks.
> > >
> > > Regarding the GLUE benchmark, we did conduct experiments and observed that our method did not significantly outperform existing baselines. It is important to note that the baseline performance on GLUE is already very high because the tasks are not particularly challenging for current models. This high baseline means there is limited room for improvement, and any advancements may not be adequately reflected in GLUE scores. Therefore, GLUE may not be the most suitable benchmark to demonstrate the advantages of our method, which is designed to address complexities in generative tasks that are not captured by GLUE.
> > >
> > > Similarly, we conducted experiments on VTAB and found results akin to those on GLUE. Like GLUE, VTAB focuses on discriminative tasks, specifically in the vision domain, which differ significantly from the generative language tasks our method addresses. Our method is tailored for generative models where redundancy appears during instruction tuning, a process essential for generative large language models but not for discriminative ones.
> > >
> > > 3. Redundancy During Instruction Tuning
> > >
> > > We have identified that redundancy appears during instruction tuning, a process that is crucial for generative large language models. Discriminative models evaluated on GLUE and VTAB do not undergo instruction tuning and, therefore, do not benefit from the redundancy reduction techniques our method employs. This explains why our method yields minor improvements on GLUE and VTAB.
> > >
> > > 4. Concerns About Scoring and Communication
> > >
> > > We noticed that the score for our paper was downgraded from a higher score to 5 (19 November 2024, 20:34 AOE) before we had the chance to respond to your initial comments. Subsequently, the score was further reduced to 3 (20 November 2024, 16:03 AOE) without additional queries or discussion. This sequence of events left us confused and concerned about how to address your feedback effectively.
> > >
> > > We apologize if any of our previous communications came across as inappropriate or offensive. It was not our intention to question your integrity or professionalism. Our earlier remarks were motivated by our confusion regarding the changes in the evaluation of our paper without further interaction or verification. We deeply respect the peer review process and value open, fair, and constructive dialogue.
> > >
> > > We are committed to addressing any concerns you may have and are eager to improve our work based on your feedback. If there are specific aspects of our paper that you believe need further clarification or revision, we are more than willing to address them comprehensively.
> > >
> > > Conclusion
> > >
> > > We believe our method makes a significant contribution to the field of generative large language models and is thoroughly evaluated using appropriate and established benchmarks. While we understand the desire for comprehensive evaluations, we feel that expecting our method to outperform on benchmarks like GLUE and VTAB, which are not designed to assess generative capabilities, does not accurately reflect its strengths or contributions.
> > >
> > > We hope this response clarifies the scope and effectiveness of our proposed method and addresses your concerns. We sincerely appreciate your time and thoughtful review, and we welcome any further questions or suggestions you may have.

---

> > > > ### Comment · Reviewer_FpRJ · 2024-11-23
> > > >
> > > > Thanks for your response.
> > > >
> > > > There was a misunderstanding on my part. I initially held some subjective biases regarding LLM tasks, which were not entirely accurate. After carefully examining the metrics of these datasets, I now agree with the authors' conclusions, and my concerns have been adequately addressed.
> > > >
> > > > Regarding the communication issues, I apologize for what appeared to be an arbitrary change in scores. Initially, the downgrade to a score of 5 stemmed from my dissatisfaction with the performance on GLUE. The subsequent downgrade was due to my skepticism regarding the reasoning for not conducting experiments on VTAB. The downgrade reason was stated in revised weakness 3, which might have been ignored by the authors and caused their confusion.
> > > >
> > > > I have raised my score back to 6. Thank you for clarifying everything and apologize for my misunderstanding again.

---

> > > > > ### Author Response · Authors · 2024-11-24
> > > > > **Thanks for your response**
> > > > >
> > > > > We are delighted to see that we have addressed your issues and questions and the score was improved!
> > > > >
> > > > > We sincerely apologize if any of our previous communications were perceived as inappropriate or offensive. It was never our intention to question your integrity or professionalism.
> > > > >
> > > > > Thank you for your valuable review comments and we apologize for any possibly inappropriate or offensive expressions during our response.

---

### Official Review · Reviewer_Qj9z · 2024-10-31

**Soundness:** 3
**Presentation:** 3
**Contribution:** 3
**Rating:** 8
**Confidence:** 4

**Summary:**

The paper introduces a novel fine-tuning methodology named NORM, aimed at reducing noise and parameter redundancy in the fine-tuning of large language models. Central to this approach is the Parameter Redundancies Fine-Tuning framework, which optimizes the parameter efficiency of LLM fine-tuning by minimizing redundancies. NORM advances this framework by employing a technique that decomposes parameters using random singular value decomposition and selects the most significant components through a Sim-Search method, which evaluates subspace similarity with the base model weights. This method has demonstrated superior performance over traditional techniques across various downstream tasks, such as instruction tuning, mathematical reasoning, and code generation, offering a significant advancement in the field of LLM fine-tuning.

**Strengths:**

The paper on NORM (Noise reduction with Reserved Majority) and the PREFT framework stands out for its originality, quality, clarity, and significance. It introduces a novel methodology combining random singular value decomposition (SVD) with Sim-Search for selective parameter retention, offering a fresh perspective on fine-tuning large language models. The research is rigorously validated across various tasks, demonstrating its robustness and relevance. Structurally, the paper excels in clarity through well-organized sections and informative visuals, making complex concepts accessible and reproducible. Significantly, NORM addresses the critical challenge of efficiently tuning large models, potentially influencing future research in machine learning optimization strategies, thereby marking a noteworthy contribution to the field.

**Weaknesses:**

1.The paper lacks comprehensive comparisons with the latest advancements in parameter-efficient fine-tuning methods, which could more precisely establish NORM's standing in the research landscape.
2.There is insufficient analysis of the computational demands and efficiency of NORM, including comparisons of resource usage and execution time against other fine-tuning methods.
3.The paper does not discuss the potential long-term effects of applying NORM in continuous learning environments where models undergo multiple fine-tuning cycles.
4.There is no detailed exploration of how NORM's performance is influenced by the choice of hyperparameters, such as the ranks used in SVD, which could affect its robustness and ease of optimization.

**Questions:**

1.Can the authors provide a more detailed analysis of the computational resources required for implementing NORM compared to traditional fine-tuning methods? This could help in assessing the practicality of NORM in resource-constrained environments.
2.The method's performance may depend on the selection of specific hyperparameters, such as the rank in SVD. Could the authors discuss how sensitive NORM is to these parameters and possibly include a sensitivity analysis or guidelines for selecting optimal hyperparameters?
3.Since the paper does not discuss long-term impacts, could the authors provide insights or preliminary results on how NORM performs in continuous learning setups or multiple iterations of fine-tuning? Understanding the stability and performance over time could significantly add to the method's robustness.
4.Are there recent advancements or other fine-tuning methodologies that were considered but not included in the comparison studies? If so, what was the reason for their exclusion, and could incorporating comparisons with these methods change the perception of NORM’s effectiveness?

---

> ### Author Response · Authors · 2024-11-17
> **Response to reviewer**
>
> Thank you for your insightful reviews and comments! We are glad to hear that you find the proposed methodology useful and innovative. It is also encouraging to see that you find our experiments extensive, the analysis insightful and the explanations realistic.
>
> > detailed analysis of the computational resources
>
> **Our response**: We illustrate the computational resources from training and evaluation aspects:
> - Training: NoRM follows the same training paradigm with LoRA tuning, so inside a normal weight multiplication process, where $X=WX+BAX$, the computational budget is $O(ndr+nd'r)$ where $d$ is the input dimension, $d'$ is the output dimension and $r$ is the LoRA rank. This training cost is the same as LoRA tuning.
> - Evaluation: NoRM uses Sim-Search to determine the most suitable adaptive rank for each parameter. This process can be conducted using any GPU or even CPU, as this process does not involve gradient update. So we exclude the requirement of computational resources during this process. After that, NoRM can merge back the denoised delta weights as LoRA tuning does: $W=W_0+B'A'$. Therefore, during inference, NoRM holds the same computational requirements as LoRA and any pre-trained language models.
>
> In conclusion, NoRM is the same efficient as LoRA tuning, and it is quite suitable for application in resource-constrain scenarios.
>
> > Sensitivity of NORM
>
> **Our response**: The only hyperparameters in NoRM lies in the search step $\tau$ and search start $s$ of the Sim-Search phase. In Figure 6 in Appendix E.1, we plot the sensitivity of choosing these two hyperparameters. This experiment shows that a small search step and a low search start bring large search scope and are prone to obtain the most suitable rank value. So to balance the search time and search performance, we choose $\tau=0.1$ and $s=1$ for all experiments.
>
> > NoRM in Continual Learning
>
> Following the experimental setup of OLoRA [1], we adopt the specified training order for continuous learning and evaluate our approach on the StandardCL benchmark. We use LLama3 8B as the base model. The training orders are as follows:
> - Order 1: dbpedia → amazon → yahoo → ag
> - Order 2: dbpedia → amazon → ag → yahoo
> - Order 3: yahoo → amazon → ag → dbpedia
> This configuration ensures consistency with prior work and allows for a fair comparison of performance across different orderings.
> We adopt Accuracy as the evaluation metric and show the performance below:
>
> | Model | Order1 | Order2 | Order3 | Average |
> |-------|--------|--------|--------|---------|
> | LoRA  | 65.34  | 74.56  | 70.44  | 70.11   |
> | NoRM  | **78.88**  | **80.08**  | **78.76**  | **79.24**   |
>
> We observe a substantial improvement over standard LoRA tuning. This enhancement stems from NoRM's ability to preserve the LoRA components most aligned with the pre-trained weights, effectively mitigating the issue of catastrophic forgetting. By prioritizing these key components, NoRM ensures better retention of prior knowledge throughout the continuous learning process.
>
> > Compare with other PEFT baselines
>
> **Our response**: We have incorporated DoRA, MoRA, and LoRA+ as recent PEFT baselines, all of which were released shortly before the submission of this paper. Here we included two more PEFT variants: LoRA-pro, which modifies LoRA's training mechanism, and PiSSA, which introduces changes to LoRA's parameter initialization. These methods have been evaluated using the TULU V2 training sets and the seven evaluation sets utilized in our study, as detailed below:
>
> | Method    | BBH   | MMLU  | TydiQA | CQA   | TruthfulQA | GSM8K | Logiqa en | Avg    |
> |-----------|-------|-------|--------|-------|------------|-------|-----------|--------|
> | LoraPro   | 37.31 | 61.72 | 45.21  | 74.77 | 47.49      | 63.38 | 33.79     | 51.95  |
> | PiSSA     | 33.16 | 59.15 | 44.37  | 75.92 | 48.71      | 69.98 | 38.71     | 52.86  |
> | NoRM      | 43.11 | 64.61 | 46.21  | 77.72 | 54.10      | 77.71 | 43.47     | 58.13  |
>
> Compared to the two other recent PEFT baselines, NoRM consistently delivers the best performance, further demonstrating its effectiveness.
>
> ## References
> [1] Xiao Wang, Tianze Chen, Qiming Ge, Han Xia, Rong Bao, Rui Zheng, Qi Zhang, Tao Gui, and Xuanjing Huang. 2023. Orthogonal Subspace Learning for Language Model Continual Learning. In Findings of the Association for Computational Linguistics: EMNLP 2023, pages 10658–10671, Singapore. Association for Computational Linguistics.

---

> ### Author Response · Authors · 2024-11-25
> **Thanks for valuable review comments**
>
> Dear reviewer
>
> I hope this message finds you well. I am writing to follow up on the rebuttal process for our paper titled “Fine-tuning with Reserved Majority for Noise Reduction” (Paper ID: 2918).
>
> First and foremost, we would like to extend our sincere gratitude to you for your time and effort in reviewing our paper. We truly value your insights and appreciate the feedback you provided, which has been instrumental in strengthening our work.
>
> We understand that the rebuttal period is to be concluded, and we would greatly appreciate your thoughts on the revisions and responses we have made to address your comments and concerns. Specifically, we hope that our clarifications have addressed any outstanding questions you may have had.
>
> If there is anything further you would like us to elaborate on, or if you require additional information, we would be more than happy to provide it.
>
> Thank you once again for your valuable contribution to the review process, and we look forward to your feedback.

---

> ### Author Response · Authors · 2024-11-27
>
> We hope this message finds you well. We are writing to kindly follow up on our previous correspondence regarding the rebuttal for our manuscript. We understand that you have a busy schedule, but we would greatly appreciate any feedback or responses you may have regarding our paper.
>
> We are eager to address any further concerns to help improve the paper. Please let us know if there is anything further we can provide.
>
> Thank you once again for your time and consideration.
>
> Best regards,
>
> Authors of Submission 2918

---

> ### Author Response · Authors · 2024-12-02
>
> Thank you once again for your detailed comments and suggestions. As the rebuttal period is approaching its end, we would greatly appreciate your feedback on whether our responses and revision have addressed your concerns. We are also happy to engage in further discussions if needed.

---

### Official Review · Reviewer_etk1 · 2024-11-09

**Soundness:** 4
**Presentation:** 3
**Contribution:** 3
**Rating:** 8
**Confidence:** 3

**Summary:**

The paper introduces a comprehensive method to address the issue in LoRA, where redundant parameters in trained weights can lead to hallucinations in Large Language Models (LLMs). In pilot experiments, the authors investigated whether dropping weights could yield performance gains across different ranks and dropping strategies, providing evidence that redundant parameters induced by LoRA can indeed degrade performance. To tackle this problem, the authors propose a new fine-tuning framework called "Parameter Redundancies Fine-Tuning," which excludes redundant parameters when merging LoRA weights with the base model. They introduce an inter-shearing method named "Noise Reduction with Reserved Majority," which decomposes LoRA weight updates using Random SVD and preserves their main components based on a novel Sim-Search method. This method aims to maximize the subspace similarity between delta LoRA weights and the original weights of the model.

**Strengths:**

The paper provides ample illustrations to present experiment results and the method pipeline. The authors conducted extensive experiments, demonstrating that the proposed NoRM method generalizes well across various models and datasets. The ablation study further supports the plausibility of searching for the rank based on subspace similarity. Since the paper introduces an improvement for merging the widely-used LoRA method with model weights, the method can significantly benefit individuals who need to train Large Language Models (LLMs) with LoRA.

**Weaknesses:**

Some equations in the paper deviate from standard notation. For instance, in lines 188 to 190, what follows "such that" is typically a statement (e.g., "we choose $\Delta W$ such that $\Delta W = \ldots$), but in the paper, it is an "argmax" result. Additionally, in lines 297 to 298, there appears to be a set-builder notation, but $\tau \cdot s, (\tau + 1) \cdot s, \ldots, r$ does not seem to be a predicate over $c$.

**Questions:**

In the "weakness" section, I pointed out some non-standard mathematical notations. Are these abbreviations or special notations? An explanation of the mathematical notations would be appreciated.

---

> ### Author Response · Authors · 2024-11-17
> **Response to reviewer**
>
> Thank you for your review and important questions! We are glad to hear that the reviewer found our proposed NoRM innovative and robust. Below we address the concerns and questions raised in the review.
>
> > L188-L190 confusion on 'such that '
>
> **Our response**: Thank you for pointing this out. We understand that the phrase "such that" may have caused some ambiguity. In this context, it was intended to specify the condition under which the new delta weight is created. To address this, we have revised the phrase to "under the condition that" in L188 in the updated manuscript. We believe this change enhances the clarity and precision of our explanation.
>
> > L297 set notation confusion
>
> **Our Response**: We appreciate your observation regarding the set notation. Our goal was to indicate that the value of $c$ is selected from the set $\\{\tau\cdot s,(\tau+1)\cdot s, \cdots, r\\}$. We acknowledge that the original notation may have been unclear. In the revised manuscript, we have corrected the set expression in Line 297 to more accurately reflect this selection. We believe this adjustment enhances the readability and precision of our mathematical notation.
>
> > I pointed out some non-standard mathematical notations. Are these abbreviations or special notations? An explanation of the mathematical notations would be appreciated.
>
> **Our response**: Thank you for bringing this to our attention. We apologize for the use of non-standard notations, which may have led to confusion. We have thoroughly reviewed the manuscript and updated the mathematical notations to align with standard conventions. Specifically, we have made revisions in Lines 188 and 297 to address these issues. Additionally, we have included explanations for any specialized notations to ensure clarity.

---

> > ### Comment · Reviewer_etk1 · 2024-12-03
> >
> > Thank you for your detailed explanation! Following a thorough review of your revised manuscript, I am pleased to report that the previously identified weaknesses have been effectively addressed. Consequently, I am delighted to bump my rating accordingly.

---

> > > ### Author Response · Authors · 2024-12-03
> > >
> > > Thank you very much for your thorough review and valuable comments, which have been instrumental in refining our paper. We are also glad to see that our paper can be revised following your suggestions! It is truly our luck to discuss with such a meticulous and knowledgeable reviewer as you! We are deeply grateful!

---

> ### Author Response · Authors · 2024-11-25
> **Thanks for valuable review comments**
>
> Dear reviewer
>
> I hope this message finds you well. I am writing to follow up on the rebuttal process for our paper titled “Fine-tuning with Reserved Majority for Noise Reduction” (Paper ID: 2918).
>
> First and foremost, we would like to extend our sincere gratitude to you for your time and effort in reviewing our paper. We truly value your insights and appreciate the feedback you provided, which has been instrumental in strengthening our work.
>
> We understand that the rebuttal period is to be concluded, and we would greatly appreciate your thoughts on the revisions and responses we have made to address your comments and concerns. Specifically, we hope that our clarifications have addressed any outstanding questions you may have had.
>
> If there is anything further you would like us to elaborate on, or if you require additional information, we would be more than happy to provide it.
>
> Thank you once again for your valuable contribution to the review process, and we look forward to your feedback.

---

> ### Author Response · Authors · 2024-11-27
>
> We hope this message finds you well. We are writing to kindly follow up on our previous correspondence regarding the rebuttal for our manuscript. We understand that you have a busy schedule, but we would greatly appreciate any feedback or responses you may have regarding our paper.
>
> We are eager to address any further concerns to help improve the paper. Please let us know if there is anything further we can provide.
>
> Thank you once again for your time and consideration.
>
> Best regards,
>
> Authors of Submission 2918

---

> ### Author Response · Authors · 2024-12-02
>
> Thank you once again for your detailed comments and suggestions. As the rebuttal period is approaching its end, we would greatly appreciate your feedback on whether our responses and revision have addressed your concerns. We are also happy to engage in further discussions if needed.

---

### Official Review · Reviewer_qqme · 2024-11-09

**Soundness:** 4
**Presentation:** 4
**Contribution:** 3
**Rating:** 8
**Confidence:** 4

**Summary:**

This paper observes the phenomenon of hallucination caused by the large redundancy in PEFT (Parameter-Efficient Fine-Tuning) and attempts to reduce the redundancy in PEFT parameters from the perspective of rank through random singular value decomposition. It then designs a SIM-SEARCH method to ensure that the selected parameter subspace is as close as possible to the distribution of pretrained parameters, thus avoiding parameters that could damage the model's capability.

**Strengths:**

1. The writing of this paper is particularly good; the story is clearly told, without over-claiming its contributions, and the logic is very clear.
2. All designs in this paper are centered around a clear goal — reducing redundant PEFT parameters that have large differences from the network distribution. Although this insight did not originally come from this paper, the paper conducted meaningful preliminary experiments to validate it.
3. The experiments in this paper achieved highly competitive results across different tasks. In some tasks where LoRA underperformed, the method in this paper surpassed full-rank fine-tuning, which is not easy to achieve.

**Weaknesses:**

1. It would be better with more mathematical explanations and analyses. How do the authors view the source of this redundancy in PEFT?
2. Why not design a smaller or uneven rank during training rather than search for it afterward? Or, is there a way to make it more end-to-end?

**Questions:**

Does increasing the rank to achieve better accuracy hold significant value for your work? Why is it specifically emphasized in such an important place like Figure 1?

---

> ### Author Response · Authors · 2024-11-17
> **Response to reviewer**
>
> Thank you very much for your detailed reviews, comments, and suggestions. We are glad to hear that you find NoRM a valuable method and that the experiments are sufficient to validate the method. Below we address the concerns and questions raised in the review.
> > It would be better with more mathematical explanations and analyses. How do the authors view the source of this redundancy in PEFT?
>
> **Our response**:
> 1. Enhancing Mathematical Explanations and Analyses:
> We appreciate the reviewer’s suggestion to include more detailed mathematical explanations and analyses. Recognizing the importance of a robust theoretical foundation, we acknowledge that our current manuscript primarily focuses on empirical results. While we have explored several potential mathematical frameworks to elucidate the redundancy observed in Parameter-Efficient Fine-Tuning (PEFT), we have not yet identified a comprehensive explanation that fully captures the phenomenon. We are committed to addressing this gap in our future work by developing a more rigorous mathematical analysis that will provide deeper insights into the sources and mechanisms of redundancy in PEFT.
> 2. Source of Redundancy in PEFT:
> We hypothesize that the redundancy in PEFT arises from the interaction between the pre-trained model’s dominant components and the additional components introduced during fine-tuning. Specifically:
> - **Major Components**: These correspond to the dominant singular values of the model's parameter matrices and primarily capture knowledge that aligns closely with the representations learned during pre-training [1]. As a result, they embody the core knowledge embedded in the pre-trained model.
> - **Minor Components**: These encode information related to the new distribution introduced by the fine-tuning corpus. They adapt the model to the specific nuances of the downstream task but may overlap with the pre-trained knowledge, leading to redundancy.
>
> This redundancy can be conceptualized as a distribution gap between the major components (pre-trained knowledge) and the minor components (downstream task-specific knowledge). Our proposed method, NoRM, mitigates this redundancy by identifying and preserving the components that are most similar to the pre-trained weights. By doing so, NoRM ensures that the fine-tuned model retains the most relevant downstream knowledge that is consistent with the pre-trained model, thereby reducing unnecessary redundancy and enhancing the efficiency of the fine-tuning process. This property also prompts NoRM to outperform LoRA in continual learning (See our response to Reviewer Qj9z).
>
> > Why not design a smaller or uneven rank during training rather than search for it afterward? Or, is there a way to make it more end-to-end?
>
> **Our response**: NoRM is designed to extract denoised components from the fine-tuned LoRA parameters by aligning them closely with the pre-trained parameters. To assign a smaller or uneven rank during training, one would need a priori knowledge of the parameter similarities, which can only be obtained post-training. This limitation makes a pre-training rank assignment infeasible within the current framework. However, we acknowledge the potential of a more end-to-end approach where rank selection could adapt dynamically during training. Exploring such a direction would require substantial methodological innovation and is an exciting avenue for future work.
>
> > Does increasing the rank to achieve better accuracy hold significant value for your work? Why is it specifically emphasized in such an important place like Figure 1?
>
> **Our response**: Figure 1 is crucial because it illustrates a key limitation of vanilla LoRA tuning: increasing the trainable rank often does not improve performance and can even introduce noise. This observation was a primary motivation for developing NoRM. Our method addresses this issue by effectively utilizing the larger representation space provided by higher ranks while systematically reducing parameter noise. By doing so, NoRM transforms the additional trainable parameters into consistent performance gains. Therefore, the emphasis on rank in Figure 1 serves to highlight both the problem we aim to solve and the effectiveness of our proposed solution.
>
> ## References
> [1] Wang H, Li Y, Wang S, et al. Milora: Harnessing minor singular components for parameter-efficient llm finetuning[J]. arXiv preprint arXiv:2406.09044, 2024.

---

> ### Author Response · Authors · 2024-11-25
> **Thanks for valuable review comments**
>
> Dear reviewer
>
> We hope this message finds you well. We are writing to follow up on the rebuttal process for our paper titled “Fine-tuning with Reserved Majority for Noise Reduction” (Paper ID: 2918).
>
> First and foremost, we would like to extend our sincere gratitude to you for your time and effort in reviewing our paper. We truly value your insights and appreciate the feedback you provided, which has been instrumental in strengthening our work.
>
> We understand that the rebuttal period is to be concluded, and we would greatly appreciate your thoughts on the revisions and responses we have made to address your comments and concerns. Specifically, we hope that our clarifications have addressed any outstanding questions you may have had.
>
> If there is anything further you would like us to elaborate on, or if you require additional information, we would be more than happy to provide it.
>
> Thank you once again for your valuable contribution to the review process, and we look forward to your feedback.

---

> ### Author Response · Authors · 2024-11-27
>
> We hope this message finds you well. We are writing to kindly follow up on our previous correspondence regarding the rebuttal for our manuscript. We understand that you have a busy schedule, but we would greatly appreciate any feedback or responses you may have regarding our paper.
>
> We are eager to address any further concerns to help improve the paper. Please let us know if there is anything further we can provide.
>
> Thank you once again for your time and consideration.
>
> Best regards,
>
> Authors of Submission 2918

---

> ### Author Response · Authors · 2024-12-02
>
> Thank you once again for your detailed comments and suggestions. As the rebuttal period is approaching its end, we would greatly appreciate your feedback on whether our responses and revision have addressed your concerns. We are also happy to engage in further discussions if needed.

---

### Author Response · Authors · 2024-11-21
**Thanks for valuable review**

Dear Reviewers,

We hope this message finds you well. We sincerely thank all reviewers for their detailed feedback and constructive comments. We are pleased to note the positive reception of our work:

1. Clarity: Reviewers qqme, Qj9z, and etk1 acknowledged that our paper is clear and easy to read.
2. Significant Performance Improvements: Reviewers qqme, Qj9z, and etk1 highlighted the substantial gains achieved by our approach.
3. Rigorous Validation: Reviewers qqme, Qj9z, and etk1 appreciated the thorough and extensive experiments, validating the robustness of our proposed NoRM.
4. Research Impact: Reviewer Qj9z commended the novel design of PreFT and NoRM, emphasizing their contributions to optimization strategies in machine learning.

In this work, we introduce PreFT and NoRM, which together establish a novel fine-tuning paradigm by addressing parameter redundancy in LoRA-based fine-tuning. Below, we summarize the key contributions of our paper:

1. A Novel Perspective on LoRA Fine-Tuning:
We revisited LoRA-based Fine-Tuning by focusing on the critical challenge of parameter redundancy. Our proposed framework, PreFT, introduces a paradigm shift in Parameter-Efficient Fine-Tuning (PEFT) by systematically improving upon existing methods.
2. The Introduction of NoRM for Redundancy Reduction:
We propose NoRM, an adaptive mechanism that retains the most relevant components of pre-trained models, effectively reducing redundancy. NoRM mitigates the problem of catastrophic forgetting (as discussed in our response to Reviewer Qj9z) and achieves notable improvements over existing PEFT methods.
3. Comprehensive Validation Across Diverse Tasks and Models:
We conducted extensive experiments with NoRM across various downstream tasks and three backbone LLMs. The results consistently demonstrate the superiority of our method over state-of-the-art PEFT and other PreFT approaches, affirming the effectiveness of addressing parameter redundancy.

We kindly inquire if it would be possible to engage in further discussion regarding our work. In our rebuttal, we have provided detailed responses to your questions and concerns. We believe that a collaborative discussion could offer additional clarity and facilitate a balanced and fair assessment of our contributions.

We understand and deeply appreciate the demands on your time, and we are truly grateful for your thoughtful consideration of this request. We look forward to your valuable feedback.

Best regards,

Authors of Paper

---

### Meta-Review · Area_Chair_SbFS · 2024-12-13

**Metareview:**

This study addresses the hallucination caused by the redundancy during LoRA finetuning. The paper is well-organized with good presentation. The proposed method is well-explained and supported by extensive empirical evaluations. Some reviewers raise concerns regarding the experimental comparison and benchmark. The authors address most of them during the rebuttal process. The AC agrees with Reviewer Qj9z and suggests that the authors add more discussions and comparisons with the latest PEFT methods, particularly the ones that also adopt SVD, e.g. AdaLoRA, PiSSA, CorDA … in their final version. In general, this is a nice work, so the AC recommends accept.

**Additional Comments On Reviewer Discussion:**

The main concerns from reviewers are the lack of comparisons with the latest studies (from Reviewer Qj9z) and the evaluation on discriminative benchmarks of GLUE and VTAB (from Reviewer FpRJ). The authors address them by adding experiments and explaining the evaluation methods of the benchmarks.

---

### Decision · Program_Chairs · 2025-01-22

Accept (Spotlight)